# Siglec-6 as a therapeutic target for cell migration and adhesion in chronic lymphocytic leukemia

Jessica Nunes[1,2], Rakeb Tafesse[1,2], Charlene Mao[1], Matthew Purcell [1], Xiaokui Mo [3], Liwen Zhang [4], Meixiao Long[1,5], Matthew G. Cyr[6], Christoph Rader[6] & Natarajan Muthusamy [1,5] ✉

Siglec-6 is a lectin receptor with restricted expression in the placenta, mast cells and memory B-cells. Although Siglec-6 is expressed in patients with chronic lymphocytic leukemia (CLL), its pathophysiological role has not been elucidated. We describe here a role for Siglec-6 in migration and adhesion of CLL B cells to CLL- bone marrow stromal cells (BMSCs) in vitro and compromised migration to bone marrow and spleen in vivo. Mass spectrometry analysis revealed interaction of Siglec-6 with DOCK8, a guanine nucleotide exchange factor. Stimulation of MEC1-002 CLL cells with a Siglec-6 ligand, sTn, results in Cdc42 activation, WASP protein recruitment and F-actin polymerization, which are all associated with cell migration. Therapeutically, a Siglec-6/CD3-bispecific T-cell-recruiting antibody (T-biAb) improves overall survival in an immunocompetent mouse model and eliminates CLL cells in a patient derived xenograft model. Our findings thus reveal a migratory role for Siglec-6 in CLL, which can be therapeutically targeted using a Siglec-6 specific T-biAb.

Sialic acid-binding immunoglobulin-like lectins (Siglecs) are known to promote cell-cell interactions and regulation of innate and adaptive immune function via glycan recognition[1]. The CD33-related (CD33r) subgroup of siglecs is mainly expressed on human immune cells like monocytes, macrophages, and a subset of B and NK cells[2]. Siglec-6 is a CD33r siglec with restricted expression in the placenta[3], mast cells[4], and tissue-like memory B cells, but is not expressed in naïve B cells[5]. It was found to be overexpressed on leukemic B cells from chronic lymphocytic leukemia (CLL) patients (B-CLL cells) when compared to healthy donor-derived B cells[6,7], making it an exciting target for CLL immunotherapy while sparing naïve B cells. A wide range of studies has explored the roles played by siglecs in tumor immunosurveillance, which makes them attractive anti-cancer targets[8]. Previous studies have reported that Siglec-6 promotes proliferation and invasion in

gestational trophoblastic disease[9,10]. While Siglec-6 has been targeted using CAR-T cells in acute myeloid leukemia (AML) and CLL[11,12], little is understood about the function of Siglec-6 in CLL.

CLL is the most prevalent adult leukemia in the Western world, with around 4000 deaths reported in the United States every year[13,14]. It is characterized by the accumulation of malignant B-CLL cells in peripheral blood from where they migrate to pro-survival niches like the bone marrow (BM) and secondary lymphoid tissues (tumor microenvironment/TME) using adhesion molecules on their surface[15]. The BM stroma protects B-CLL cells from apoptosis whereas the lymph node microenvironment stimulates B cell receptor signaling and tumor proliferation in CLL[16,17]. Various chemokines including CCL21 and CXCL12 are involved in B cell migration, but they are not CLL-specific and are thus not ideal therapeutic targets[15,18]. ROR1 is another

[1]Comprehensive Cancer Center, The Ohio State University, Columbus, OH, USA. [2]Molecular, Cellular and Developmental Biology Graduate Program, The Ohio State University, Columbus, OH, USA. [3]Center for Biostatistics, The Ohio State University, Columbus, OH, USA. [4]Campus Chemical Instrument Center, The Ohio State University, Columbus, OH, USA. [5]Division of Hematology, Department of Internal Medicine, The Ohio State University, Columbus, OH, USA. [6]UF Scripps Biomedical Research, University of Florida, Jupiter, FL, USA. ✉e-mail: raj.muthusamy@osumc.edu

CLL-specific receptor that promotes CLL cell migration via Wnt5a signaling and has been validated as a therapeutic target to treat CLL patients[19–21]. However, the complex crosstalk between B-CLL cells and TME as well as the underlying mechanism via which these adhesion molecules promote migration and adhesion is still not completely understood[17].

Sialic acids are negatively charged sugar units present as capping molecules on the glycan surface of glycoproteins and glycolipids[22,23]. Sialic acid residues are expressed on human bone marrow-derived stromal cells (BMSCs)[24], and previous studies have shown that sialylated ligands on bone marrow sinusoidal endothelium promote CD22/Siglec-2 dependent homing of B cells to the bone marrow[25]. While in vitro studies have shown that Siglec-6 binds to sialyl Tn (sTn), a truncated O-glycan consisting of sialic acid attached to GalNAc α-O-Serine/Threonine[3,26,27], the role of sTn in Siglec-6 mediated migration and adhesion of B-CLL cells is not clear.

In this work, we use a proteomics approach to investigate the mechanistic role of Siglec-6 in migration and adhesion of B-CLL cells. We show that the ligand of Siglec-6 (sTn) is overexpressed on CLL patient-derived BMSCs and our studies suggest that sTn stimulation leads to Cdc42 activation and actin polymerization via a Siglec-6-DOCK8-dependent axis. Using a Siglec-6/CD3-bispecific T cell-recruiting antibody, we demonstrate improved overall survival in an immunocompetent mouse model and elimination of CLL cells in a patient-derived xenograft model, thereby paving the way for pre-clinical evaluation of Siglec-6 targeted therapeutics.

## Results

### Siglec-6 is overexpressed on B cells from CLL patients

Flow cytometry analysis of CD19+ CD5+ B cells from CLL patients revealed varying levels of surface expression of Siglec-6 ranging from 2.45 to 749.8 median fluorescence intensity (MFI) ($n = 31$, mean MFI 103.13). In contrast, CD19+ B cells from healthy donors revealed minimal Siglec-6 expression ranging from 1.11 to 32.84 MFI ($n = 10$, mean MFI: 11.9) (Fig. 1a). Expression of Siglec-6 on B cells from CLL patients was also confirmed by confocal microscopy (Fig. 1a). Evaluation of Siglec-6 expression among patients based on differences in IGHV mutational status and 13q deletion failed to reveal any association with Siglec-6 expression (Fig. 1b). Overall, Siglec-6 is a B-CLL-specific marker with limited expression on normal B cells.

### Siglec-6 ligand sialyl Tn (sTn) is expressed on bone marrow mesenchymal stromal cells (BMSCs) from CLL patients but not healthy donors

The bone marrow is one of the major sites that B-CLL cells home to in CLL patients[28,29]. Sialyl Tn (sTn) is a tumor-associated carbohydrate antigen that is associated with poor prognosis and tumor metastasis in colon and breast cancer[26,30]and has been shown to exhibit the strongest binding affinity for Siglec-6 compared to other siglec ligands like sialyllactose and Tn[1,27,31]. To determine sTn expression on BMSCs, we derived BMSCs from bone marrow aspirates of CLL patients and from hip bone samples of healthy donors. Flow cytometric analysis of BMSCs identified by CD73 and CD90 expression revealed significantly increased surface expression of sTn on BMSCs from CLL patients (mean MFI 38,000) when compared to normal donors (mean MFI 2000) (Fig. 1c). Further, western blotting analysis also revealed sTn overexpression (with a specific band detected at ~150 kDa) in CLL-BMSCs (Fig. 1d).

To determine if Siglec-6 interacts with sTn, we used the DT-40 cell line to test the binding of sTn to Siglec-6 since DT-40 cells are devoid of any surface siglec molecules. DT-40 cells transfected with a CMV-Siglec-6 cDNA expression construct revealed surface expression of Siglec-6 detected by biotinylated human anti-human Siglec-6 monoclonal antibody (mAb) JML-1 (a CLL patient-derived mAb[7]), analyzed by flow cytometry (Supplementary Fig. 1a). Biotinylated sTn (sTn-B)

displayed increased binding to Siglec-6+ DT-40 cells when compared to empty vector-transfected cells, which was inhibited by JML-1 antibody (Ab) (9.78-fold reduction) indicating that Siglec-6 binds to sTn (Fig. 2a). We then used an ELISA based assay wherein Protein A coated plates were coated with IgG Fc (negative control) or a Siglec-6 Fc fusion protein, followed by probing with sTn-B and colorimetric quantification. STn displayed a dose-dependent increase in binding to Siglec-6 Fc when compared to IgG Fc negative control (Fig. 2b). Next, to determine if sTn expressed on CLL-BMSCs is the ligand that is mediating binding to Siglec-6, we performed immunoprecipitation with Siglec-6 Fc fusion protein on CLL and normal donor BMSCs followed by immunoblotting with sTn antibody. We found sTn expression in CLL-BMSCs subjected to Siglec-6 Fc pull-down (with a specific band detected at ~150 kDa), with limited sTn protein detected in normal donor BMSCs, indicating that sTn on the surface of CLL-BMSCs is indeed interacting with Siglec-6 (Fig. 2c). To further confirm binding specificity, flow cytometry analysis via Siglec-6 Fc binding and detection using biotinylated JML-1 on CLL-BMSCs was performed. We observed a 1.6-fold decrease in Siglec-6 binding upon blocking the BMSCs with sTn Ab when compared to isotype control, though this decrease was not significant (Fig. 2d), indicating the presence of other ligands of Siglec-6 that were not blocked by the sTn Ab. Interestingly, sialidase treatment significantly abrogated Siglec-6 Fc binding (800-fold decrease) when compared to isotype control, indicating Siglec-6 dependency on sialic acids for binding to sTn (Fig. 2d). Overall, Siglec-6 displays specificity to the ligand sTn.

### Siglec-6 associates with DOCK8, a guanine nucleotide exchange factor

Previous reports studying the role of Siglec-6 in trophoblastic cell invasion[9,10] led us to hypothesize a potential role for Siglec-6 and its associated proteins in CLL cell migration and adhesion. To systematically identify Siglec-6 interacting proteins, we resorted to an unbiased proteomics approach with mass spectrometry. Since the MEC1 CLL cell line, which is widely used as a cell model system for CLL[32], displays low levels of surface Siglec-6 expression, we used the MEC1-002 cell line that was derived from the MEC1 cell line by enriching for Siglec-6+ population[7]. Increased Siglec-6 expression on the surface of MEC1-002 compared to MEC1 cells is shown in Supplementary Fig. 1b. Protein lysates from MEC1-002 cells were subjected to pull-down with Dynabeads coupled with either a commercial immunoprecipitation validated anti-Siglec-6 Ab (LifeSpan BioSciences) or IgG1 isotype Ab, followed by nano-liquid chromatography-nanospray tandem mass spectrometry. Our results identified several candidate proteins including DOCK8, a guanine nucleotide exchange factor (GEF), which showed the most significant enrichment with Siglec-6 pull-down (2.9-log2 fold change) compared to the isotype control Ab pull-down (Fig. 3a). A list of candidate proteins with log2 fold change >1.5 is shown in supplementary Table 1. To independently verify that DOCK8 interacts with Siglec-6, protein lysates from MEC1-002 or primary CD19+ CD5+ B-CLL cells were immunoprecipitated with the same Siglec-6 Ab that was used for the mass spectrometry analysis. Analysis of co-immunoprecipitated proteins with anti-DOCK8 antibody by western blot confirmed the interaction of Siglec-6 with DOCK8 (Fig. 3b).

### Siglec-6 promotes migration and adhesion of CLL cells in vitro

CLL-specific overexpression of Siglec-6 led us to hypothesize a potential role for Siglec-6 in the migration of B-CLL cells to the tumor microenvironmental (TME) niche. MEC1-002 cells stained either with human IgG1 isotype Ab or with anti-Siglec-6 JML-1 Ab, followed by subsequent staining with a PE-conjugated commercial Siglec-6 Ab showed that blocking with JML-1 Ab but not isotype Ab prevented subsequent binding of PE-conjugated Siglec-6 Ab, indicating that JML-1 specifically binds to Siglec-6 and serves as a blocking antibody

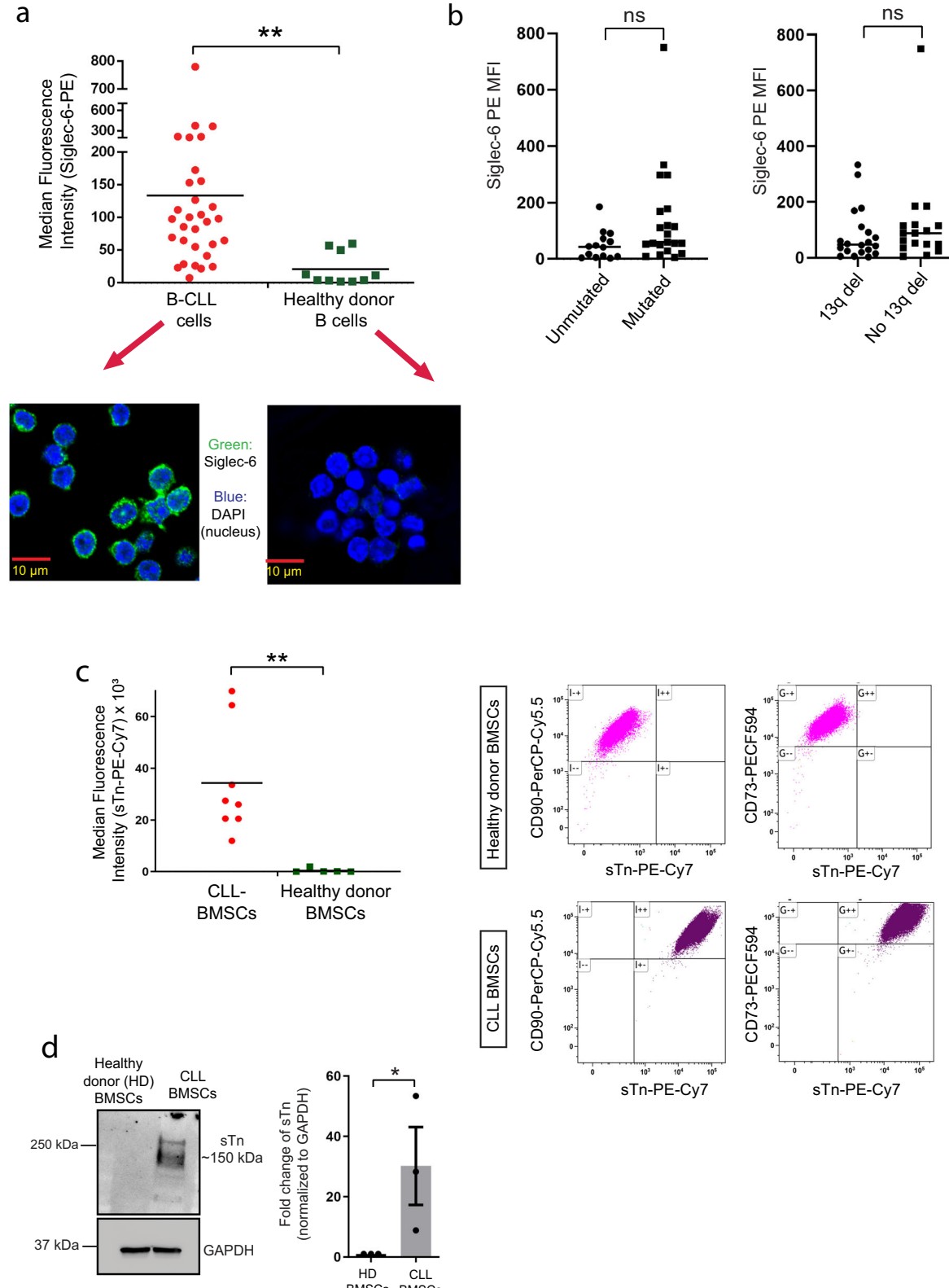

(Supplementary Fig. 1c). Consistent with this, the JML-1 Ab blocked attachment of Siglec-6+ primary B-CLL cells to sTn+ CLL-BMSCs in an in vitro co-culture assay (2.35-fold decrease) (Fig. 4a), indicating the need for Siglec-6 in attachment of B-CLL cells to CLL-BMSCs. Importantly, the JML-1 Ab had no effect on attachment of Siglec-6⁻ normal donor B cells to CLL-BMSCs (Fig. 4b). The JML-1 Ab also had no effect on attachment of B-CLL cells to normal BMSCs which lack Siglec-6

ligand sTn (Fig. 4c). Further, CLL-BMSCs promoted in vitro migration of B-CLL cells in a transwell migration assay ($n = 5$, 6.7-fold increase), which was in turn inhibited by the JML-1 Ab (4-fold reduction) (Fig. 4d). Thus, JML-1 Ab could block both in vitro migration as well as adhesion of B-CLL cells to CLL-BMSCs. The JML-1 Ab could also negate in vitro migration of Siglec-6⁺ MEC1-002 cells (4.2-fold reduction) (Supplementary Fig. 1d). A comparison between in vitro migration of MEC1-

**Fig. 1 | Siglec-6 and its ligand, sialyl Tn (sTn), are overexpressed in chronic lymphocytic leukemia (CLL) patients when compared to healthy donors. a** Flow cytometry analysis showing median fluorescence intensity (MFI) normalized to isotype control of surface Siglec-6 expression on B cells from CLL patients (31 patients) is significantly higher when compared to normal B cells (10 donors) [$t(39) = 4.48$, $P = 0.0004$]. Representative immunofluorescence images showing surface Siglec-6 expression on B-CLL cells and not on normal B cells are shown below. **b** CLL patient samples from (**a**) were segregated based on different genetic aberrations: IgVH mutation status (mutated or unmutated) or 13q deletion status (13q del). **c** Flow cytometry analysis showing MFI normalized to isotype control of surface sTn expression on bone marrow stromal cells (BMSCs) from 8 CLL patients (CLL-BMSCs) is significantly higher when compared to BMSCs from normal healthy donors (normal BMSCs) (5 donors) [$t(11) = 8.69$, $P = 0.001$]. Representative dot plots showing sTn overexpression on CD73$^+$ CD90$^+$ CLL-BMSCs are shown on the right. Gates were set based on 'fluorescence minus one' controls. **d** Western blotting analysis and ImageJ quantification showing sTn expression (~150 kDa) on CLL but not healthy donor BMSCs [$t(4) = -4.5$, $P = 0.01$] (3 independent donors each). Graphs show the mean ± standard error of the mean. MFI values were obtained by subtracting from isotype control. *$P < 0.05$; **$P < 0.005$ by paired two-tailed $t$-test (**a**–**d**). ns: not significant. Scale bar for (**a**) is 10 μm. Source data are provided in a source data file.

002 WT cells and primary CLL cells with or without Siglec-6 blockade with JML-1 Ab is shown in supplementary Fig. 2a. To test if biological inhibition of Siglec-6 is sufficient to inhibit migration of MEC1-002 cells, we used the CRISPR-Cas9 technique to establish stable knockout of *SIGLEC-6* in MEC1-002 cells (Fig. 4e, Supplementary Fig. 2d, and Supplementary Fig. 4b). Consistent with the observations in primary CLL cells, knocking out *SIGLEC-6* in MEC1-002 cells resulted in significant reduction in migration towards sTn+ CLL-BMSCs (2.5-fold reduction) (Fig. 4e). These results thus suggest a migratory and adhesion role of Siglec-6 in B-CLL cells.

Next, we examined the expression of Siglec-6 in CD49d+ CLL cells which drives the homing of CLL cells to lymphoid tissues[15,33]. Six CLL patient samples were tested, and none of them showed bimodal CD49d expression as reported by Tissino et al.[34]. Siglec-6 was expressed on CD49d+ cells, and no correlation was found between CD49d and Siglec-6 expression levels in terms of mean fluorescence intensity (MFI) ($r = -0.77$, $P = 0.08$) (Supplementary Fig. 3a). We then evaluated Siglec-6 expression on the recent stromal-niche emigrant (RSE) subset (CD5hiCXCR4lo) from the bone marrow and lymph node niche and compared it to expression on the CD5loCXCR4hi long-term circulating cells (LTCs) subset[35]. Again, no difference in Siglec-6 expression was observed between the CD5hiCXCR4lo and CD5loCXCR4hi subset (mean MFI 1133.9 vs 1079.8) (Supplementary Fig. 3b). Overall, Siglec-6 expression in CLL patients is not limited to a specific prognostic subset.

Ibrutinib is a tyrosine kinase inhibitor that is known to block the migration of CLL cells[36,37]. CLL patients undergoing ibrutinib treatment do not show altered surface Siglec-6 expression when compared to pre-ibrutinib treatment, as reported by Cyr et al.[38]. Thus, we investigated whether ibrutinib has a synergistic effect with JML-1 Ab on inhibition of CLL cell adhesion and/or migration. Ibrutinib treatment significantly inhibited migration (1.7-fold decrease) (Supplementary Fig. 2b) and adhesion (2.3-fold decrease) (Supplementary Fig. 2c) of CLL cells to CLL-BMSCs when compared to DMSO control. However, the combination of ibrutinib with JML-1 Ab did not significantly inhibit migration (Supplementary Fig. 2b) or adhesion (Supplementary Fig. 2c) of CLL cells when compared to JML-1 Ab or ibrutinib treatment alone.

### Functional role of DOCK8 and Siglec-6 in CLL cell migration

DOCK8 has been shown as an important regulator of cell migration in different cell types including T cells and dendritic cells[39,40]. To determine if DOCK8 is required for Siglec-6 dependent B-CLL migration, we used the CRISPR-Cas9 technique to establish stable knockout of *DOCK8* in MEC1-002 cells that was confirmed by western blot analysis (Supplementary Fig. 4a) and confocal microscopy (Supplementary Fig. 4b). In vitro transwell migration assays revealed significant reduction in migration of *DOCK8* knockout MEC1-002 cells towards CLL-BMSCs (2.4-fold reduction), implicating a role for DOCK8 in Siglec-6 dependent B-CLL cell migration (Fig. 5a). Consistent with this, siRNA mediated knockdown of *SIGLEC-6* in MEC1-002 *DOCK8* KO cells failed to enhance the compromised migratory capacity observed in *DOCK8* KO cells (Fig. 5a).

DOCK proteins including DOCK8 are localized to the cell membrane via phosphoinositide binding which in turn localizes GEF activity to cell membrane compartments[40], which is necessary for integrating signals from the cell membrane to regulate pathways involved in actin polymerization and subsequent cytoskeletal rearrangement[41]. To determine if surface-expressed Siglec-6 tethers DOCK8 to the cell membrane in CLL cells, we evaluated DOCK8 protein levels in the membrane fraction of MEC1-002 WT, MEC1-002 *SIGLEC-6* KO cells, Siglec-6+ primary B-CLL cells, and normal healthy donor (ND) B cells. We observed that Siglec-6 + MEC1-002 WT cells and primary B-CLL cells had higher levels of DOCK8 at the cell membrane [2-fold, $t(6) = -5.72$, $P = 0.001$; and 3-fold, $t(8) = 3.33$, $P = 0.01$, respectively] when compared to *SIGLEC-6* KO MEC1-002 cells and normal donor B cells (Fig. 5b). Further, there was no significant difference in total DOCK8 levels in the whole cell lysate protein fraction of MEC1-002 WT, MEC1-002 *SIGLEC-6* KO, B-CLL and normal donor B cells (Fig. 5c), indicating that the higher membrane levels of DOCK8 observed in MEC1-002 WT and B-CLL cells was due to increased association of Siglec-6 with DOCK8. Importantly, engagement of Siglec-6 with its ligand sTn resulted in increased DOCK8 recruitment at the cell membrane in MEC1-002 WT cells and B-CLL cells [2-fold, $t(6) = 2.8$, $P = 0.03$; and 2-fold, $t(8) = 4.89$, $P = 0.001$, respectively] (Fig. 5b). This was not seen in the *SIGLEC-6* KO and normal donor B cells, indicating a ligand-mediated Siglec-6 dependent role in DOCK8 recruitment to the membrane in MEC1-002 cells.

To identify which cell compartment is involved in the localization of DOCK8 to the cell membrane, we also evaluated DOCK8 protein levels in the cytoplasmic and nuclear fraction of each cell type. Interestingly, while the cytoplasmic fraction displayed no changes in DOCK8 protein levels (Supplementary Fig. 5a), the nuclear fraction of both MEC1-002 *SIGLEC-6* KO and normal donor B cells had higher levels of DOCK8 [2.7-fold, $t(4) = 3.3$, $P = 0.03$; and 3-fold, $t(8) = -3.77$, $P = 0.005$, respectively] when compared to MEC1-002 WT and B-CLL cells respectively (Supplementary Fig. 5b), indicating that DOCK8 may be localized to cell membrane from the nucleus in Siglec-6 + MEC1-002 WT and B-CLL cells.

### Sialyl Tn/Siglec-6/DOCK8 axis-dependent activation of Cdc42 results in actin polymerization in CLL cells

DOCK8 has been previously implicated in T cell and dendritic cell migration via activation of Cdc42, and Arp2/3 complex-driven actin polymerization[39,42], but no studies have interrogated Siglec-6/DOCK8 axis-dependent Cdc42 activation in B-CLL cells. MEC1-002 cells with functional Siglec-6 and DOCK8 displayed enhanced Cdc42 activation upon stimulation with sTn (as evidenced by increased Cdc42-GTP levels), which was not observed in *SIGLEC-6* and *DOCK8* knockout cells, indicating that sTn signals via Siglec-6 and DOCK8 to activate Cdc42 (Fig. 6a, b). Further, JML-1 anti-Siglec-6 Ab that blocks sTn binding to Siglec-6 (Fig. 1c) inhibited the sTn mediated Cdc42 activation in MEC1-002 WT cells (Fig. 6a, b), confirming that sTn mediated Cdc42 activation is dependent on sTn-Siglec-6 interaction. Active GTP-bound Cdc42 binds to WASP protein resulting in WASP activation which in turn promotes actin polymerization[43–45]. We observed that sTn-mediated Cdc42 activation also resulted in the recruitment of WASP

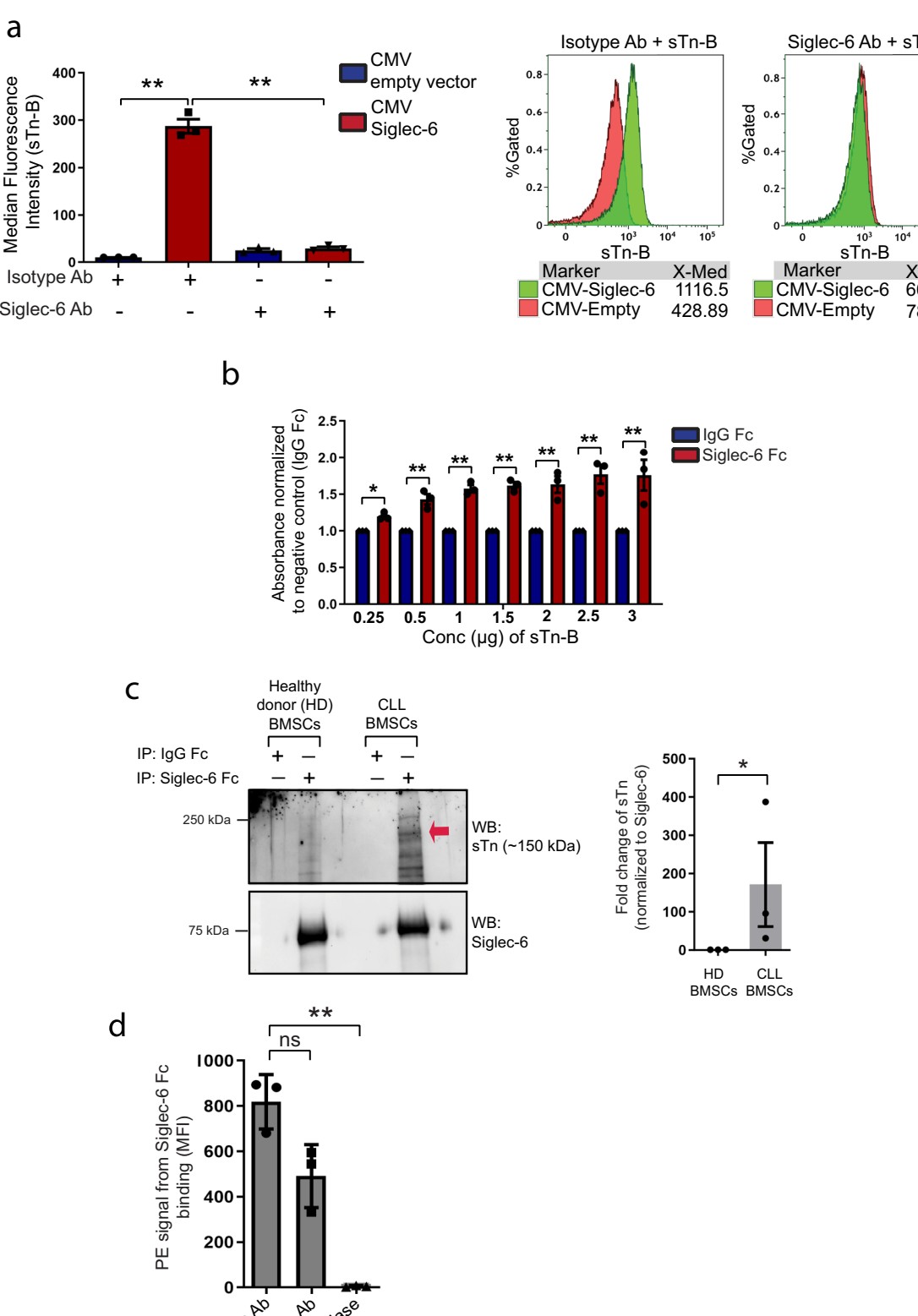

protein, which was not seen in MEC1-002 WT cells blocked with JML-1 Ab, nor in MEC1-002 *SIGLEC-6* or *DOCK8* KO cells (Fig. 6a, b). Importantly, we evaluated the effect on actin polymerization in MEC1-002 WT, *SIGLEC-6* or *DOCK8* KO cells stimulated with sTn for 10 min, followed by staining with phalloidin to detect and visualize filamentous actin filaments (F-actin) via confocal microscopy. WT MEC1-002 cells displayed ~2-fold enhanced F-actin polymerization in response to stimulation with sTn when compared to unstimulated cells. Moreover, MEC1-002 cells lacking Siglec-6 or DOCK8 showed compromised F-actin polymerization upon stimulation with sTn (Fig. 6c, d). Overall, these results show the mechanistic pathway involved in Siglec-6-dependent migration wherein sTn signals via the Siglec-6-DOCK8 axis to promote Cdc42 activation, WASP protein recruitment, and subsequent actin polymerization.

**Fig. 2 | Siglec-6 interacts with its ligand, sTn. a** In vitro assay showing JML-1 (Siglec-6 Ab) significantly inhibits binding of biotinylated sTn (sTn-B) to Siglec-6 ($P = 0.0000015$) via flow cytometry analysis. Representative overlay plots from one replicate are shown below. **b** ELISA-based assay showed binding of sTn to Siglec-6 via colorimetric analysis, with maximum binding at 3 μg sTn ($P = 5.46 \times 10^{-9}$). **c** Co-immunoprecipitation via pull-down with Siglec-6 Fc or non-specific IgG Fc and western blotting probing with sTn Ab shows sTn+ proteins in CLL-BMSCs [$t(4) = -4.45$, $P = 0.01$]. Representative blot shown from one of 3 independent patient samples. Quantification was done by normalizing to Siglec-6 using ImageJ analysis. **d** In vitro assay testing binding of Siglec-6 to sTn on CLL-BMSCs. Siglec-6 binding to CLL-BMSCs in isotype Ab treated cells was compared to sTn Ab treated cells [$t(4) = 1.54$, $P = 0.19$] and sialidase treated cells [$t(4) = 14.89$, $P = 0.0001$]. Binding of Siglec-6 Fc was measured using biotinylated JML-1 Ab via flow cytometry analysis. **a, b** $n = 3$ independent experiments. **c, d** 3 independent donors. Graphs show the mean ± standard error of the mean. *$P < 0.05$; **$P < 0.005$ by paired two-tailed $t$-test (**c**) two-sided mixed effect modeling (**a, b, d**). ns not significant. Ab antibody. Source data are provided in the source data file.

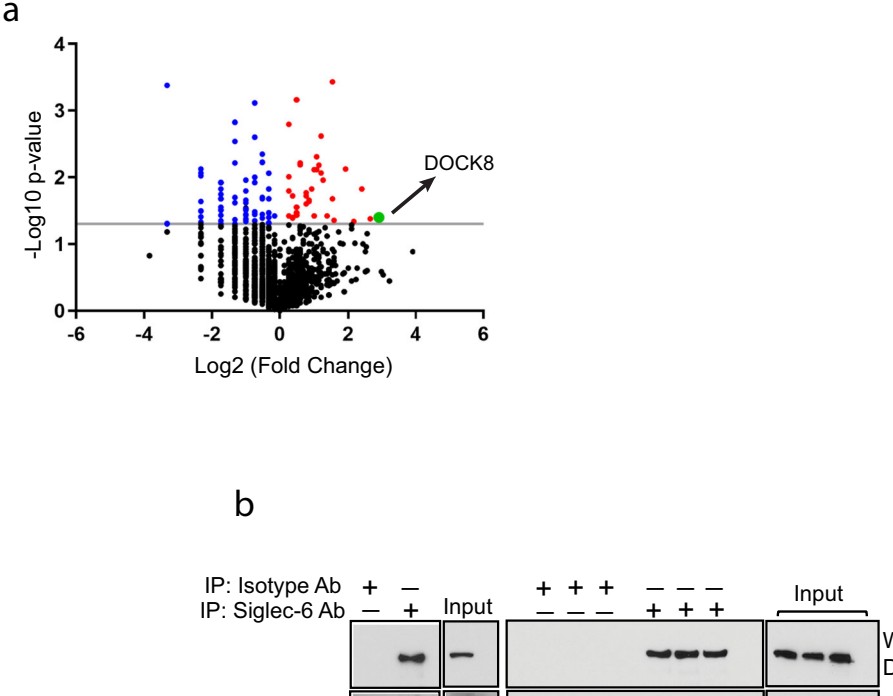

**Fig. 3 | Siglec-6 associates with DOCK8, a guanine nucleotide exchange factor. a** Mass spectrometric analysis to identify Siglec-6 interacting partners. Volcano plot depicts a log2 fold change of proteins associated with Siglec-6 normalized to isotype control. −log10 $P$-value > 1.3 was the significance threshold used, calculated using a two-tailed student $t$-test. Green dot indicates DOCK8 which had the highest fold change (2.9−log2 fold, $P = 0.04$). Black dots indicate non-significant proteins. Blue dots indicate significant proteins that were enriched with isotype Ab pull-down and are thus not associated with Siglec-6. Red dots indicate candidate proteins that associate with Siglec-6. Data pooled from 3 independent experiments. **b** Co-immunoprecipitation via pull-down with Siglec-6 Ab and western blotting probing with DOCK8 Ab confirmed Siglec-6-DOCK8 interaction in MEC1-002 and B-CLL cells from 3 independent patient samples. Ab antibody. A full list of the proteomics dataset is provided in the supplementary information file. Source data are provided in the source data file.

Siglec-6 was previously implicated as a signaling molecule in human trophoblasts via recruitment of the phosphatase SHP-2[46]. We next tested if Siglec-6 mediated triggering of the DOCK8/WASP axis is dependent on SHP-2. SiRNA-mediated knockdown of *SHP-2* in MEC1-002 cells did not abrogate ligand-mediated Cdc42 activation, WASP recruitment (Supplementary Figs. 6a–c), and actin polymerization (Supplementary Fig 7), when compared to MEC1-002 cells transfected with non-targeting siRNA, indicating that Siglec-6 is signaling through DOCK8 through a pathway that is independent of SHP-2.

**JML-1 antibody inhibits homing of Siglec-6 positive MEC1-002 and primary B-CLL cells to the spleen and bone marrow in vivo**
Malignant B-CLL cells are characterized by homing to the spleen and bone marrow (BM), followed by retention within these tissues that in turn provide favorable conditions that enable CLL survival[13,14,47]. Thus, we next investigated whether migration of MEC1-002 cells to the spleen and BM of NSG mice could also be impaired by anti-Siglec-6 targeted antibodies. Evaluation of the ability of MEC1-002 cells to migrate to primary BMSCs from C57Bl/6 mice in an in vitro set-up revealed inhibition of migration of Siglec-6 + MEC1-002 (2.9-fold decrease) (Supplementary Fig. 8a) in the presence of the JML-1 Ab. We next evaluated Siglec-6 dependency in vivo migration in the NSG mouse model. MEC1-002 cells or primary B-CLL cells treated with JML-1 Ab or isotype control, were injected intravenously via tail vein in NSG mice and the in vivo migration of the cells was followed by flow cytometry post-engraftment. Analysis of MEC1-002 cells in the spleen and BM of the recipient mice 24 h post-engraftment, revealed a significant reduction in the number of MEC1-002 cells with JML-1 Ab

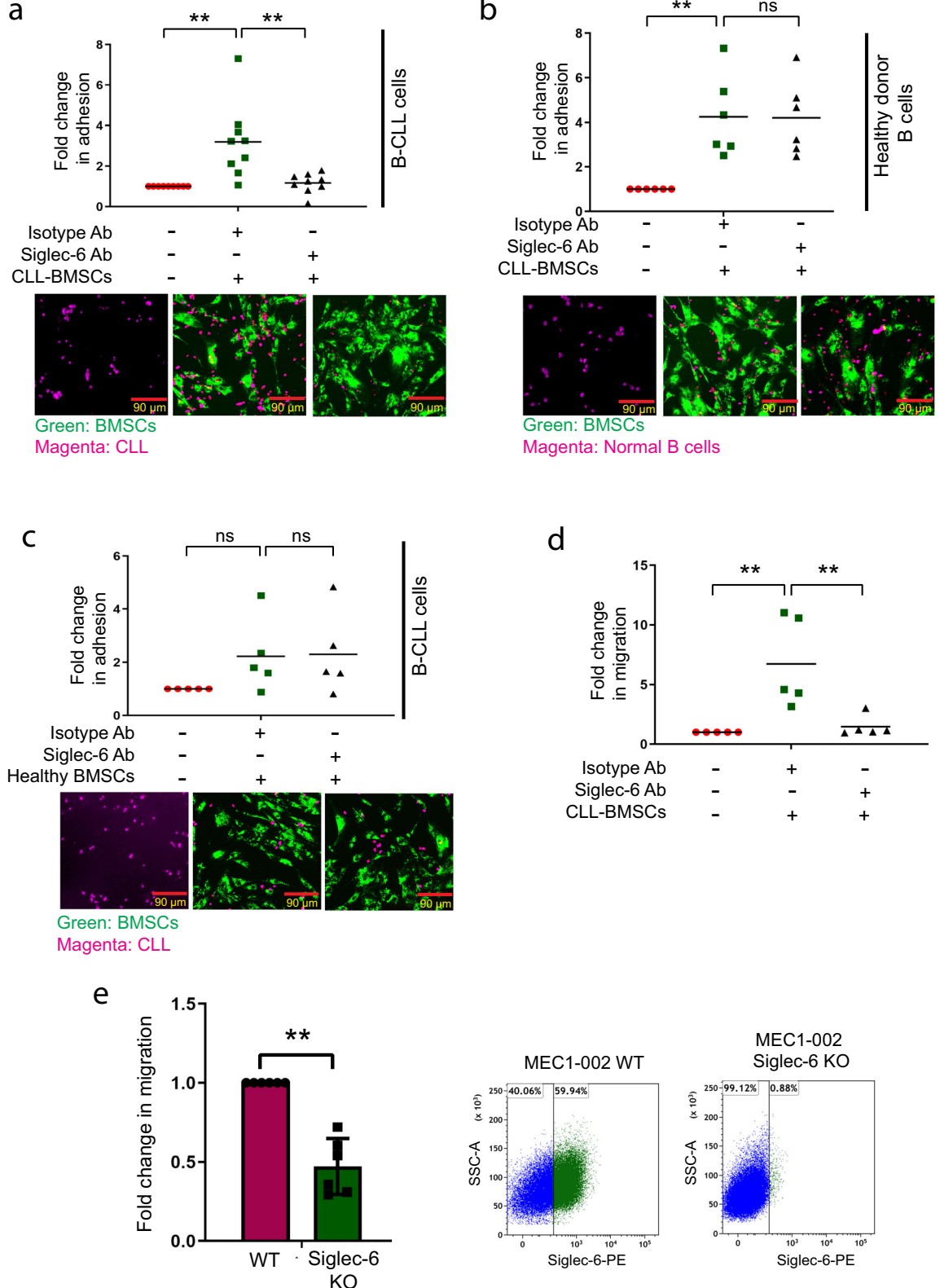

Green: BMSCs
Magenta: CLL

Green: BMSCs
Magenta: Normal B cells

Green: BMSCs
Magenta: CLL

blocking compared to isotype control (5.4-fold vs 3.5-fold) (Fig. 7a). Similarly, the number of B-CLL cells detected in the spleen and BM after JML-1 Ab blocking was also significantly reduced when compared to isotype control Ab (10-fold vs 5-fold) (Fig. 7b). Taken together, these results show that the JML-1 Ab decreased homing of malignant B cells to the spleen and BM.

## Efficacy of Siglec-6 targeted T-biAb against Siglec-6 + CLL cells ex vivo

To evaluate the Siglec-6 targeting Siglec-6/CD3-bispecific T cell-recruiting antibody (T-biAb) in an immunocompetent mouse model, we required a model to accurately recapitulate the characteristics of human CLL while also expressing the human Siglec-6 (hSiglec-6)

**Fig. 4 | Siglec-6 promotes migration and adhesion of chronic lymphocytic leukemia (CLL) cells in vitro. a, b** In vitro adhesion assay comparing adhesion of B-CLL cells (9 patients) (**a**) or healthy donor B cells (6 donors) (**b**) to CLL- bone marrow stromal cells (BMSCs). The JML-1 monoclonal antibody (Siglec-6 Ab) significantly reduced the adhesion of B-CLL cells when compared to isotype control [$t(16) = 4.43$, $P = 0.0004$]. **c** JML-1 did not affect the attachment of B-CLL cells to normal BMSCs (5 patients). **d** Transwell migration assay shows that the JML-1 inhibits in vitro migration of B-CLL cells towards CLL-BMSCs when compared to blocking with the isotype Ab [$t(8) = 5.10$, $P = 0.0009$] (5 patients). **e** Transwell migration assay shows that in vitro migration of MEC1-002 wild-type (WT) cells towards CLL-BMSCs is significantly reduced when compared to MEC1-002 Siglec-6 knock-out (KO) cells [$t(5) = 5.3$, $P = 0.003$], n = 6 biological replicates. Representative dot plots from one biological replicate show loss of surface Siglec-6 expression on MEC1-002 Siglec-6 KO cells after CRISPR/Cas9 knock-out of Siglec-6. CountBright absolute counting beads were used to count the number of cells that have migrated or attached. For (**a–c**), representative co-culture assays were imaged using the ECHO Revolve R4 microscope. Graphs show the mean ± standard error of the mean. *$P < 0.05$; **$P < 0.005$ by two-sided mixed effect modeling (**a–d**) or paired two-tailed $t$-test (**e**). ns not significant. Ab antibody. Source data are provided in the source data file.

target protein. To do this, we generated hSiglec-6 transgenic mice (hSiglec-6-Tg) and crossed them with the Eμ-TCL1 mouse, an established model of IgVH unmutated CLL that has been extensively used to evaluate CLL therapeutics[48–50] (Fig. 8a). The hSiglec-6-Tg was created using an expression vector that contained IgVH promoter and IgH-μ enhancer elements to achieve B cell-specific hSiglec-6 expression. Presence of the transgene in two founder lines was determined by PCR (Supplementary Fig. 8b). Expression of hSiglec-6 on B cells was confirmed by flow cytometry performed on peripheral blood samples, with majority of the B cells expressing Siglec-6 (76.22% in a representative mouse) (Supplementary Fig. 8c). Further, expression of hSiglec-6 on CD5+ CD19+ leukemic B cells in peripheral blood of hSiglec-6×TCL1 mice was also confirmed by flow cytometry (Supplementary Fig. 8d). We evaluated the cytotoxic efficacy of the T-biAb ex vivo using the hSiglec-6 + TCL1 leukemic B cells. Human CD3 (huCD3) T cells which express the human epitope of the CD3ε chain were isolated from humanized CD3 transgenic mice[51,52] and stimulated with concanavalin A for 7 days. They were then cocultured with either B cells from CLL patients, normal donor B cells, Siglec-6⁻ TCL1 splenic B cells, or Siglec-6⁺ hSiglec-6×TCL1 splenic B cells, along with the Siglec-6 targeting T-biAb. A non-targeting CD123 × huCD3 T-biAb was used as a negative control since the target B cells do not express CD123. After 5 h in culture, significant cytotoxicity as demonstrated by annexin V/PI staining was observed in the case of B-CLL patient cells (4-fold) when compared to Siglec-6⁻ healthy donor B cells (Fig. 8b), as well as in Siglec-6⁺ hSiglec-6×TCL1 splenic B cells (2.6-fold) when compared to Siglec-6⁻ TCL splenic B cells (Fig. 8c).

We next evaluated the clinical utility of the Siglec-6 targeting T-biAb against treatment-naïve CLL patients using autologous T cells ex vivo. PBMCs from CLL patients ($n = 5$) with T cell: CLL ratios ranging from 1:0.8 to 1:25 were cultured with either Siglec-6 targeting T-biAb or CD123 T-biAb for 7 days. Significant cytotoxicity was observed with Siglec-6 T-biAb (3-fold) when compared to CD123 T-biAb (Fig. 8d). These results thus demonstrate targeted killing of Siglec-6⁺ CLL cells.

### Siglec-6 targeted T-biAb improves overall survival of immuno-competent mice with hSiglec-6×TCL1 leukemia and eliminates patient CLL cells in vivo

Following the confirmation of ex vivo activity of Siglec-6 T-biAb, we evaluated its efficacy on overall survival in vivo. To this end, 5 million sorted CD5+ CD19+ CD3-Siglec-6+ leukemic cells from hSiglec-6×TCL1 mice (Fig. 8e), were engrafted into huCD3 mice and monitored for disease. Upon CLL development (>3% circulating leukemic cells), mice were treated i.v with either 0.1 mg/kg Siglec-6 T-biAb or non-targeting CD123 T-biAb, weekly once, until the mice met early removal criteria (ERC). A trend towards delay in leukemia progression was observed in the mice treated with Siglec-6 T-biAb when compared to the CD123 T-biAb treated group (Supplementary Fig. 8e). We did not observe any signs of treatment-related toxicity or decrease in bodyweight over the course of the study (Supplementary Fig. 8f). Most importantly, Siglec-6 T-biAb treatment led to significant survival benefit (Fig. 8f), mean survival 170 days (Siglec-6 T-biAb) vs 90 days (CD123 T-biAb).

Next, we examined the efficacy of the Siglec-6 T-biAb against CLL in vivo using a patient-derived xenograft model[53], which recapitulates interactions between the patient's CLL and T cells. NSG mice were engrafted with PBMCs from treatment-naïve CLL patients via tail vein injection. The presence of Siglec-6 on CLL cells and the absence of CD123 on PBMCs was confirmed via flow cytometry (Fig. 9a). After CLL engraftment was confirmed on day 3, mice were dosed with either Siglec-6 T-biAb or CD123 T-biAb (non-targeting control) once a week for 2 weeks. One week post-treatment, mice that received the Siglec-6 T-biAb had significantly reduced circulating leukemic count when compared to mice that received the CD123 T-biAb [23-fold decrease, $t(55) = 8.2$, $P = 4.07 \times 10^{-11}$] (Fig. 9b). After two weekly injections, Siglec-6 T-biAb treated mice had significantly reduced leukemic count in the blood [17-fold decrease, $t(55) = 8.4$, $P = 2.05 \times 10^{-11}$] as well as spleen [3.8-fold, $t(13) = 3.15$, $P = 0.008$] when compared to CD123 T-biAb treated mice (Fig. 9b, c). Taken together, these data demonstrate the efficacy of Siglec-6 T-biAb against Siglec-6+ leukemic cells.

## Discussion
BTK inhibitors like ibrutinib and acalabrutinib and Bcl-2 inhibitors like venetoclax have caused a paradigm shift in the treatment of CLL. Expression of BTK, Bcl-2, CD20, and CD19 targets in diseased CLL and normal B cells leads to indiscriminate elimination of normal B cells leading to compromised B cell functions in patients that respond to therapy. Identifying tumor-specific targets and uncovering their biological role is an important therapeutic goal in CLL treatment. In the present study, we have shown that Siglec-6 promotes migration and adhesion of B-CLL cells. This is mediated through the interaction of Siglec-6 with DOCK8 in Siglec-6 ligand sTn-dependent actin polymerization via Cdc42 activation and WASP protein recruitment (Fig. 9d). Most importantly while Siglec-6 blocking antibody JML-1 prevented Siglec-6 mediated cell adhesion and migration, an anti-hSiglec-6 x anti-hCD3 T-biAb improved overall survival in immuno-competent mice engrafted with hSiglec-6 + CLL cells, and induced rapid killing of CLL cells in a CLL patient-derived xenograft model. Collectively, these studies uncover a role for Siglec-6 in sTn-mediated DOCK8-dependent migration and adhesion of B-CLL cells.

Although Siglec-6 has been implicated in trophoblastic cell invasion, as an inhibitory receptor on mast cells, and as an immune checkpoint in bladder cancer patients[4,54,55] limited studies highlight its role in CLL. Molecular interactions with the lymph node and bone marrow tumor microenvironment (TME) are essential for the survival of B-CLL cells[15,56], and trafficking of B-CLL cells between the blood and the TME is regulated by specific receptors on these B-CLL cells and their cognate ligands. STn is associated with poor prognosis in many types of epithelial cancers including gastric and colon cancer[26]. While expression of sTn has been reported in a few leukemias and lymphomas[57–59], it has not been studied in CLL. Our finding of selective overexpression of sTn, a reported ligand of Siglec-6, in CLL patient BMSCs is intriguing and has translational implications as it provides opportunities to target and block interactions between Siglec-6 and sTn on CLL-BMSCs, thereby inhibiting homing and/or retention of Siglec-6 + B-CLL cells to the bone marrow TME. Since the scope of our

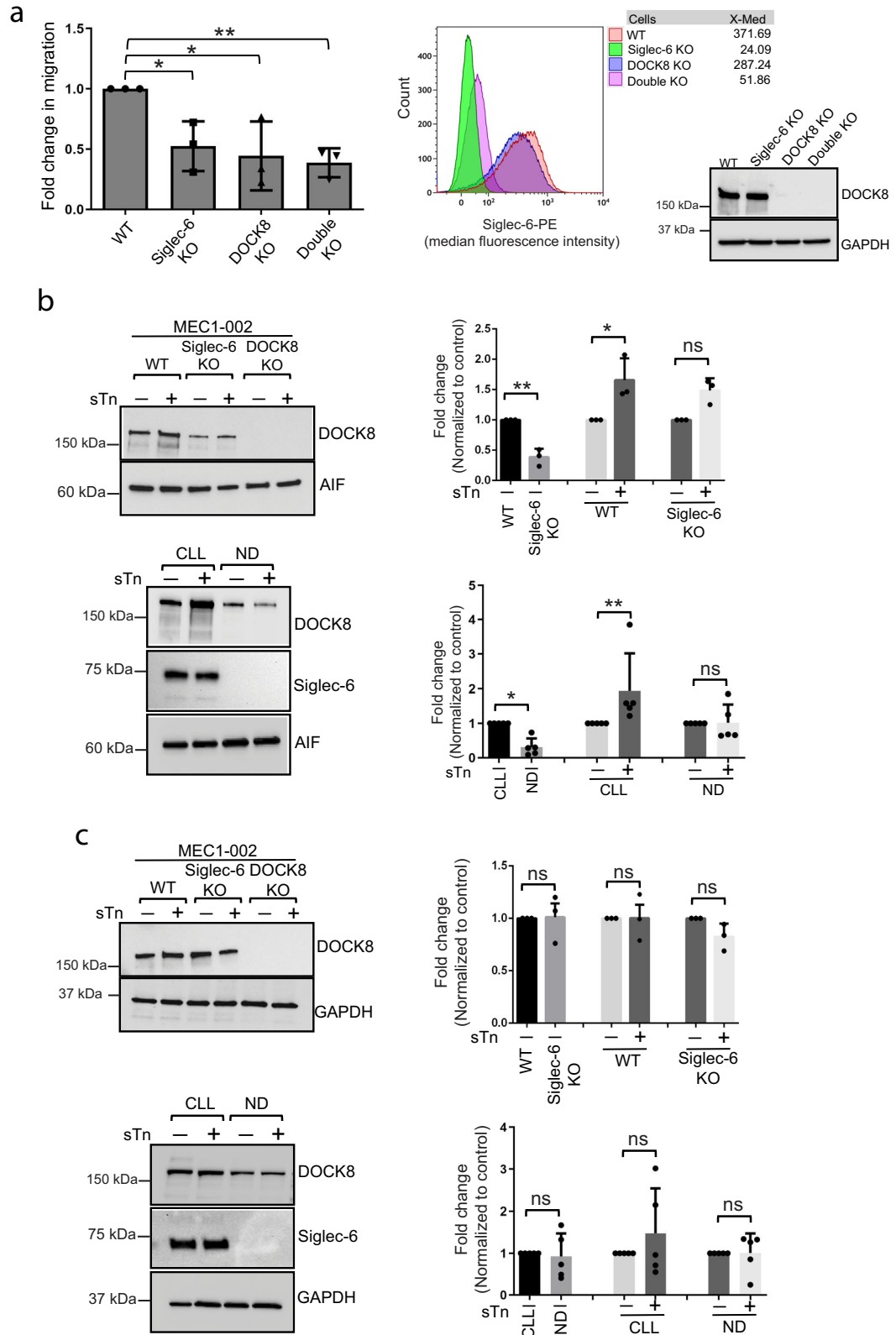

work is limited to the bone marrow niche, further studies will need to be carried out to assess Siglec-6 ligands in the lymph nodes of CLL patients.

Our mass spectrometry and biochemical data showing Siglec-6 interaction with DOCK8 is very promising as it reveals a role for Siglec-6 signaling via DOCK8. The interaction between Siglec-6 and DOCK8 has physiological implications for tethering DOCK8 to the cell membrane, where the migratory actin polymerization events occur. Indeed, we observed that MEC1-002 WT and B-CLL cells had increased relative levels of DOCK8 at the cell membrane with a concomitant decrease in nuclear DOCK8 levels when compared to MEC1-002 SIGLEC-6 KO cells and normal donor B cells, respectively, implicating Siglec-6 interaction dependent DOCK8 localization in Siglec-6+ cells. CLL cells interact with the TME via a network of adhesion molecules,

**Fig. 5 | Functional role of DOCK8 and Siglec-6 in chronic lymphocytic leukemia (CLL) cell migration. a** Transwell migration assay showing that MEC1-002 Siglec-6 knock-out (KO), MEC1-002 DOCK8 KO, and MEC1-002 double KO (with knockdown of both Siglec-6 and DOCK8) cells had significantly reduced migration towards CLL-bone marrow stromal cells (BMSCs) when compared to MEC1-002 wild-type (WT) cells [$t(6) = -3.67$, $P = 0.01$; $t(6) = -4.26$, $P = 0.005$; and $t(6) = 4.51$, $P = 0.004$, respectively]. Data from $n = 3$ biological replicates. Right panels show loss of of DOCK8 in the MEC1-002 DOCK8 KO and double KO cell lines by western blot analysis, and loss of surface Siglec-6 expression on MEC1-002 Siglec-6 KO and double KO cell lines by flow cytometry analysis, from one biological replicate.

**b**, **c** Western blotting to test membrane localization of DOCK8. Protein lysates were then collected and separated into the membrane fraction (**b**) and whole lysate (WCL) fraction (**c**). Apoptosis-inducing factor (AIF) and GAPDH were used as loading control for the membrane and WCL fraction, respectively. DOCK8 protein levels were measured by densitometric quantification of immunoblots and normalized to AIF or GAPDH loading control. Representative immunoblots are shown. Cell line data are from $n = 3$ independent experiments. For experiments with primary cells, $n = 5$ independent donors were used. Graphs show the mean ± standard error of the mean. *$P < 0.05$; **$P < 0.005$ by two-sided mixed effect modeling. ns: not significant. Source data are provided in the source data file.

cell surface ligands, chemokines, and cytokines including VCAM1, fibronectin, and CXCR4[60]. The Siglec-6/DOCK8 axis most likely represents one network that is involved in driving CLL migration to the BM niche, that could be functioning concurrently with other networks in Siglec-6 + CLL cells. Further studies to elucidate the specific domains involved in Siglec-6 interaction with DOCK8 will provide insight into how this pathway is connected to other molecular networks involved in CLL migration and adhesion.

Our ability to block the Siglec-6/DOCK8 interaction and downstream events with a blocking antibody (JML-1) has translational implications. JML-1 is a patient-derived, high-affinity anti-Siglec-6 Ab that emerged in a human allogeneic setting[6,38]. The epitope of JML-1 Ab was mapped to the N-terminal lectin domain, indicating that it can potentially interfere with siglec–glycan interactions[38]. Indeed, our data show that blocking with JML-1 Ab could hinder CLL cell migration and adhesion as well as sTn-mediated downstream effects, which reinforces the therapeutic benefit of humanized Siglec-6 targeted antibodies.

The humanized Siglec-6 + CLL mouse model described here in combination with the human CD3 transgenic mice has enabled the evaluation of Siglec-6-directed CD3-bispecific antibody. While these studies corroborate work done by Cyr et al. in cell line-derived xenograft mouse models[38], the immunocompetent mouse models described here provide opportunities to investigate in vivo mechanisms of bispecific antibody functions and explore ways to enhance and modulate their functions in an immunocompetent environment. Two groups have developed Siglec-6 targeted chimeric antigen receptor (CAR)-T cells for the treatment of CLL and AML[11,12], which demonstrates the benefits of targeting Siglec-6-expressing tumor cells with T cell-based therapies. However, several aspects favor T-biAbs over CAR-T cells, including being a low-cost off-the-shelf therapy with high accessibility to more patients and a shorter time to treatment[61]. Another important aspect of Siglec-6 targeted immunotherapies is the expression of Siglec-6 on memory B cells and mast cells[4,5]. Memory B cells play a vital role in the protection of the host from subsequent infections with the same pathogen, and the effectiveness of vaccines relies on the generation of immunological memory[62]. Thus, depletion by Siglec-6 targeted T-biAb therapy may have adverse effects on this B cell compartment. Further, mast cells play a role in IgE-mediated allergic reactions via FceRI receptor[63], and they may also be targeted by the Siglec-6 T-biAb leading to complications in mast cell activation during allergic reactions. While no major cytotoxic side effects were observed in our in vivo studies as well as in prior studies evaluating Siglec-6 targeted therapies in AML and CLL[11,12,38], further studies will need to be carried out to study the effects of Siglec-6 targeting on these hematopoietic cell populations in vivo.

In summary, the present study describes a previously unknown aspect of B-CLL migration to the bone marrow niche which involves activation of Cdc42 via a sTn/Siglec-6/DOCK8-dependent mechanism. Therapeutically, a Siglec-6 targeted T-biAb can specifically eliminate Siglec-6 + CLL cells and provide a survival benefit, thus supporting the rationale for clinical evaluation of Siglec-6 targeted therapies in patients with CLL.

## Methods

All the research carried out here complies with relevant ethical regulations. Patient samples were obtained from subjects who gave written and informed consent according to an IRB-approved protocol. All in vivo mouse experiments were performed in accordance with Federal and Institutional Animal Care and Use Committee (IACUC) requirements.

### Cell culture
The MEC1 cell line was obtained from DSMZ (ACC 497) and the DT-40 cell line was obtained from ATCC (CRL-2111). The MEC1-002 cell line was derived from the Siglec-6+ fraction of the MEC1 cell line by FACS sorting[7] and was provided by Dr. Rader (University of Florida) under a Material Transfer Agreement (MTA) and Institutional Review Board (IRB) approval. All primary cells and cell lines were cultured at 37 °C in media supplemented with heat-inactivated fetal bovine serum (FBS) and 1% penicillin/streptomycin/L-glutamine (P/S/G; Gibco). Primary PBMCs and B cells, and B cell lines (MEC1, DT-40, and MEC1-002) were cultured in a 5% CO2 incubator in RPMI with 10% FBS. Primary bone marrow stromal cells (BMSCs) were cultured in a hypoxia incubator (3% O2) in low glucose DMEM with 15% FBS. Cell lines were validated via short tandem repeat analysis at The Ohio State University Genomic Services Core. They were routinely tested for mycoplasma contamination (Universal Mycoplasma Detection Kit, ATCC 30−1012K), and were discarded after passage twenty. Detailed information is provided in the supplementary information file.

### Human samples and study approval
Peripheral blood mononuclear cells (PBMCs) and bone marrow mesenchymal stromal cells (BMSCs) were collected from normal healthy donors or CLL patients in compliance with the Declaration of Helsinki. All subjects gave written and informed consent for the blood products to be used for research according to an IRB-approved protocol. Blood from CLL patients was collected at The Ohio State University Comprehensive Cancer Center (Columbus, OH, USA). Normal healthy donor PBMCs were obtained from Red Cross partial leukocyte preparations. PBMCs were isolated using Ficoll density-gradient centrifugation, and Ficoll-Paque Plus (GE Healthcare). B cells from CLL patients were isolated using a B cell RosetteSep enrichment kit (#15064, StemCell Technologies) according to the manufacturer's protocol. CLL-BMSCs were cultured from bone marrow aspirates from CLL patients. Normal healthy donor BMSCs were isolated from the spongiform bone marrow of hip bone samples stored in the Leukemia Tissue Bank (LTB) at OSU.

### Animal studies
All animal procedures were performed in accordance with Federal and Institutional Animal Care and Use Committee (IACUC) requirements.
**Animal studies:** The human Siglec-6 transgenic mouse (hSiglec-6-Tg) was generated on a C57BL/6 background (Jackson Laboratory, 000664) at the OSUCCC Transgenic Mouse Facility by conventional methodology[64]. B cell restricted hSiglec-6 expression is driven by immunoglobulin (Ig) $V_H$ promoter and $Ig_H$-μ enhancer elements in the

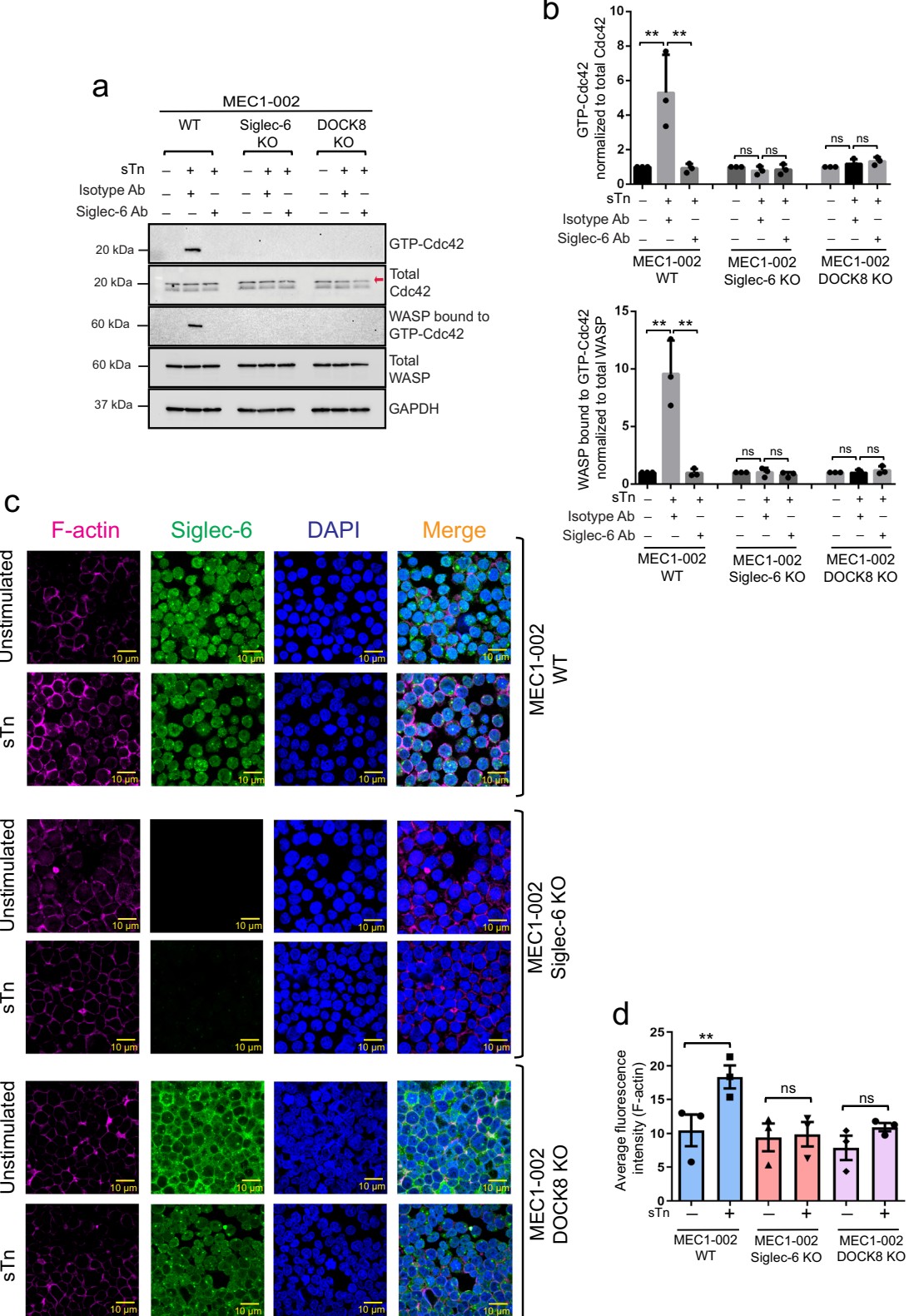

pBH expression vector[64]. The transgenic construct was generated by ligating the cDNA sequence of human Siglec-6 into BglII sites within the pBH vector via blunt end ligation. To generate our Siglec-6+ leukemia model, we crossed hSiglec-6-Tg mice with Eμ-TCL1 mice (C57BL/6 background)[48,49]. Mice were housed in microisolator cages under controlled temperature and humidity. All animal procedures were performed in accordance with Federal and Institutional Animal Care

and Use Committee (IACUC) requirements. Two founder lines were established and the presence of the transgene in the founder lines was determined by PCR. Primer sequences to detect the *SIGLEC-6* gene are as follows: Forward primer 5′-TTTGTGCATTGGAAACCAGA-3′ and reverse primer 5′-CTGTCTGGAACTGGTGCTGA-3′. Humanized CD3ε mice were kindly provided by Dr. Meixiao Long (The Ohio State University). To generate them, human CD3 transgenic mice were obtained

**Fig. 6 | Sialyl Tn promotes Siglec-6 and DOCK8-dependent Cdc42 activation, WASP protein recruitment, and actin polymerization. a** Cdc42 activation assay kit was used to pull-down GTP-bound Cdc42. GTP-Cdc42 and active WASP protein were detected using immunoblotting. Total Cdc42 band is depicted by a red arrow. **b** STn ligand stimulation of MEC1-002 wild-type (WT) cells increased GTP-Cdc42 levels ($P = 0.003$) and active WASP protein levels ($P = 0.0001$) as measured by densitometric quantification of immunoblots and normalized to total Cdc42 and total WASP protein in the whole cell protein lysate, respectively. Blocking with JML-1 significantly reduced GTP-Cdc42 levels ($P = 0.003$) and active WASP protein levels

($P = 0.0001$) in MEC1-002 WT cells. **c** Confocal immunofluorescent analysis of filamentous actin (F-actin) filaments was done by staining with phalloidin. Ligand stimulation of MEC1-002 WT resulted in a significant increase in F-actin polymerization ($P < 6.24 \times 10^{-15}$). **d** ImageJ was used to quantify fluorescence intensity from F-actin staining from 20 random fields examined over 3 independent experiments. **a**, **b** $n = 3$ independent experiments. Graphs show the mean ± standard error of the mean. *$P < 0.05$; **$P < 0.005$ by two-sided mixed effect modeling. ns: not significant. Scale bar for (**c**) is 10 µm. Source data are provided in the source data file.

from Jackson Laboratory (Stock No: 020456, B6.Cg-Tg (CD3E) 600Cpt/J). Wild-type C57/BL6 mice were bred with these mice, and mice with peripheral T cell numbers and CD4:CD8 ratios closely resembling those of wild-type littermates were chosen for subsequent breeding. 8–12 week mice that were heterozygous for the human CD3 transgene were used for subsequent studies. An equal ratio of 8-12 week male and female mice were maintained for experiments with huCD3 (B6.Cg-Tg (CD3E) 600Cpt/J) mice. For experiments with NSG mice (NOD.Cg-Prkdcscid Il2rgtm1Wjl/SzJ), 8-week male mice were used since they were more cost-effective.

## Mass spectrometry

Protein lysates were made from MEC1-002 cells followed by pull-down with anti-human Siglec-6 antibody (clone 2G6, LifeSpan Biosciences) or mouse IgG2b isotype control antibody (Invitrogen) conjugated to Dynabeads® Magnetic Beads (Invitrogen). Beads were submitted to the Ohio State University Proteomics Center at the Campus Chemical Instrument Center for proteomics analyses. Beads were washed twice with 50 mM ammonium bicarbonate (25–50 µL each time depending on bead volume). Each time, the supernatant was kept and pooled. After the second wash, 5 µL of DTT (5 µg/uL in 50 mM ammonium bicarbonate) was added and the sample was incubated at 56 °C for 15 min. After the incubation, 5 µL of iodoacetamide (15 mg/ml in 50 mM ammonium bicarbonate) was added and the sample was kept in the dark at room temperature for 30 min. Either 250 ng or 500 ng of sequencing grade-modified trypsin (Promega) prepared in 50 mM ammonium bicarbonate was added to the sample reaction was carried on at 37 °C for overnight, additional 50 mM ammonium bicarbonate was added to make the final volume of the samples to 100 µL. The reaction was quenched the next morning by adding 0.1% acetic acid for acidification. The supernatant was taken out and concentrated for LC/MSMS analysis. Nano-liquid chromatography-nanospray tandem mass spectrometry (Nano-LC/MS/MS) of protein identification was performed on a Thermo Scientific Orbitrap Fusion mass spectrometer equipped with an EASY-Spray™ Sources operated in positive ion mode, carried out at The Ohio State University Campus Chemical Instrument Center (CCIC). Capillary-liquid chromatography-nanospray tandem mass spectrometry (Capillary-LC/MS/MS) of protein identification was performed on a Thermo Scientific orbitrap Fusion mass spectrometer equipped with an EASY-Spray™ Sources operated in positive ion mode. Samples (6.4 µL) were separated on an easy spray nano column (PepmapTM RSLC, C18 3 µ 100 A, 75 µm × 150 mm Thermo Scientific) using a 2D RSLC HPLC system from Thermo Scientific. Each sample was injected into the µ-Precolumn Cartridge (Thermo Scientific) and desalted with 0.1% Formic Acid in water for 5 min. The injector port was then switched to inject, and the peptides were eluted off of the trap onto the column. Mobile phase A was 0.1% Formic Acid in water and acetonitrile (with 0.1% formic acid) was used as mobile phase B. Flow rate was set at 300 nL/min. Typically, mobile phase B was increased from 2% to 20% in 105 min and then increased from 20–32% in 10 min and again from 32–95% in 1 min and then kept at 95% for another 4 min before being brought back quickly to 2% in 1 min. The column was equilibrated at 2% of mobile phase B (or 98% A)

for 15 min before the next sample injection. MS/MS data was acquired with a spray voltage of 1.6 KV and a capillary temperature of 305 °C is used. The scan sequence of the mass spectrometer was based on the preview mode data dependent TopSpeed™ method: the analysis was programmed for a full scan recorded between m/z 375–1500 and an MS/MS scan to generate product ion spectra to determine the amino acid sequence in consecutive scans starting from the most abundant peaks in the spectrum in the next 3 s. To achieve high mass accuracy MS determination, the full scan was performed at FT mode and the resolution was set at 120,000 with internal mass calibration. The AGC Target ion number for the FT full scan was set at $4 \times 105$ ions, the maximum ion injection time was set at 50 ms, and the micro scan number was set at 1. MSn was performed using HCD in ion trap mode to ensure the highest signal intensity of MSn spectra. The HCD collision energy was set at 32%. The AGC Target ion number for the ion trap MSn scan was set at 3.0E4 ions, maximum ion injection time was set at 35 ms, and the micro scan number was set at 1. Dynamic exclusion is enabled with a repeat count of 1 within 60 s and a low mass width and high mass width of 10 ppm. Sequence information from the MS/MS data was processed by converting the. raw files into a merged file (.mgf) using MSConvert (ProteoWizard). Isotope distributions for the precursor ions of the MS/MS spectra were deconvoluted to obtain the charge states and monoisotopic m/z values of the precursor ions during the data conversion. The resulting mgf files were searched using Mascot Daemon by Matrix Science version 2.5.1 (Boston, MA), and the database was searched against the most recent Uniprot databases. The mass accuracy of the precursor ions was set to 5 ppm, and the accidental pick of 1 13 C peaks was also included in the search. The fragment mass tolerance was set to 0.5 Da. Carbamidomethylation (Cys) is used as a fixed modification and considered variable modifications were oxidation (Met) and deamidation (N and Q). Four missed cleavages for the enzyme were permitted. A decoy database was also searched to determine the false discovery rate (FDR) and peptides were filtered according at 1% FDR. Proteins identified with at least two unique peptides were considered reliable identification. Any modified peptides were manually checked for validation.

## CRISPR/Cas9 mediated knock-out

The Alt-R™ CRISPR-Cas9 System from Integrated DNA Technologies (IDT) was used to generate Siglec-6 and DOCK8 stable knock-out MEC1-002 cell lines. Pre-designed guide RNAs targeting *DOCK8* (5′-CGTCGCACAGCACGT-3′) and *SIGLEC-6* (5′-GTAGCCATAACCATA-3′) were obtained from IDT. Delivery of ribonucleoprotein (RNP) complexes into MEC1-002 cells was performed as follows using manufacturer's protocol (IDT): the RNP complex consisting of CRISPR-Cas9 cRNA, tracrRNA, and Cas9 was electroporated into MEC1-002 cells using the Amaxa Nucleofector system. 48 h post electroporation, single cells were sorted into individual wells in 96 well plates using a BD Bioscience BD FACSAria™ Fusion flow cytometer. Twenty single cell-derived clones were analyzed for DOCK8 or Siglec-6 knockdown. MEC1-002 DOCK8 KO (clone #1) and MEC1 Siglec-6 KO (clone #5) were selected for all further experiments.

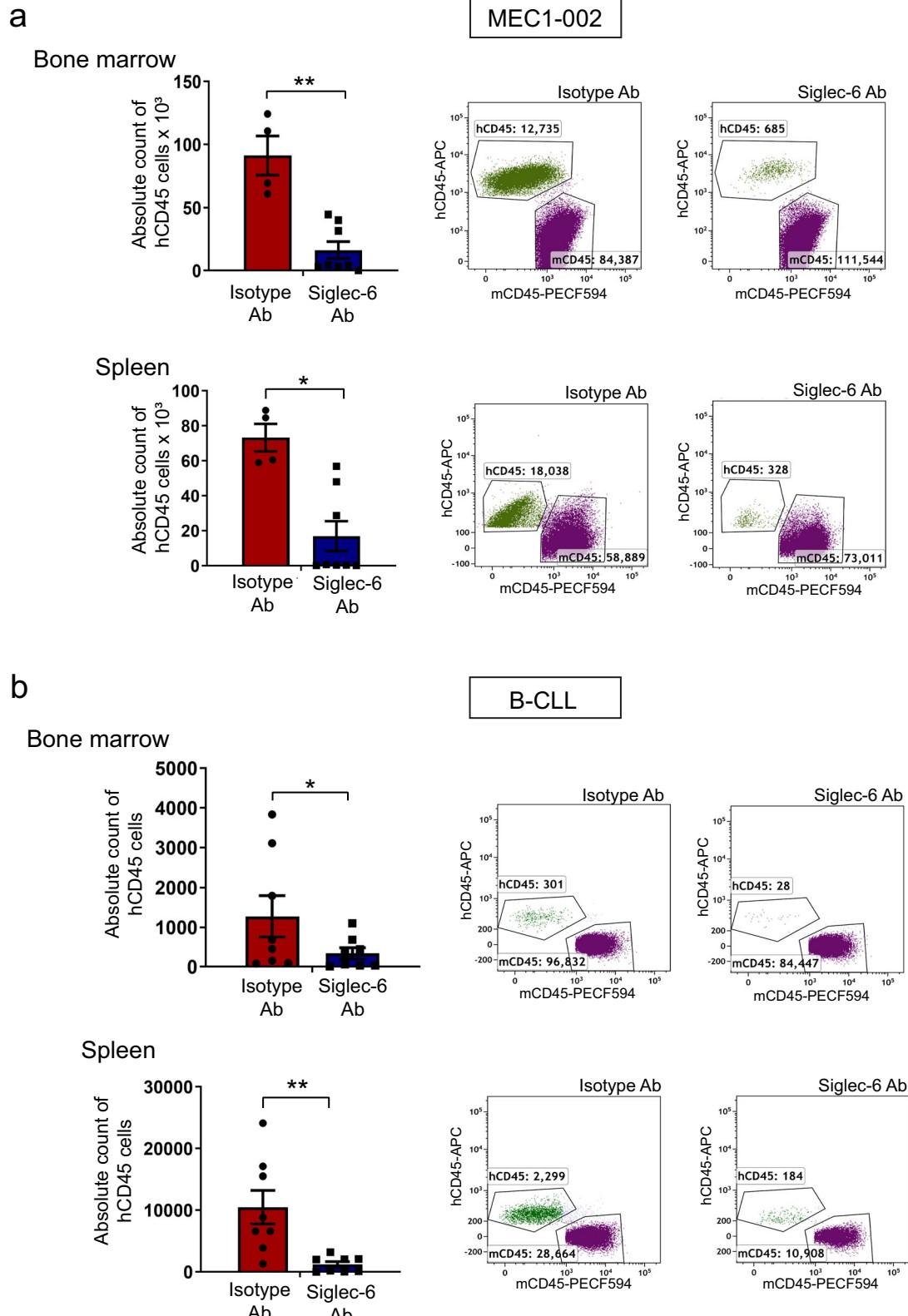

**Fig. 7 | JML-1 antibody inhibits homing of Siglec-6 positive MEC1-002 cells and B-CLL cells to the spleen and bone marrow. a** MEC1-002 cells ($1 \times 10^7$ cells) were blocked either with isotype Ab or JML-1 (Siglec-6 Ab) for 1 h, and then injected into the tail vein of biologically independent NOD.Cg-Prkdc$^{scid}$ Il2rg$^{tm1Wjl}$/SzJ (NSG) mice, $n = 4$ for isotype Ab group and $n = 8$ for JML-1 Ab group. Bone marrow and spleen were analyzed by flow cytometry for the presence of MEC1-002 cells using anti-human hCD45 antibody and counted using CountBright beads. Representative dot plots from one mouse are shown. JML-1 significantly inhibited homing of MEC1-002 cells to the spleen ($P = 0.0062$) and bone marrow ($P = 0.0043$). **b** Same as (**a**), but B-CLL cells were used, $n = 8$ for the isotype Ab group and $n = 8$ for the JML-1 Ab group. JML-1 significantly inhibited homing of primary B-CLL cells to the spleen ($P = 0.003$) and bone marrow ($P = 0.007$). 8-week male NSG mice were used. Graphs show the mean ± standard error of the mean. *$P < 0.05$; **$P < 0.005$ by two-sided mixed effect modeling. ns not significant. Ab antibody. Source data are provided in the source data file.

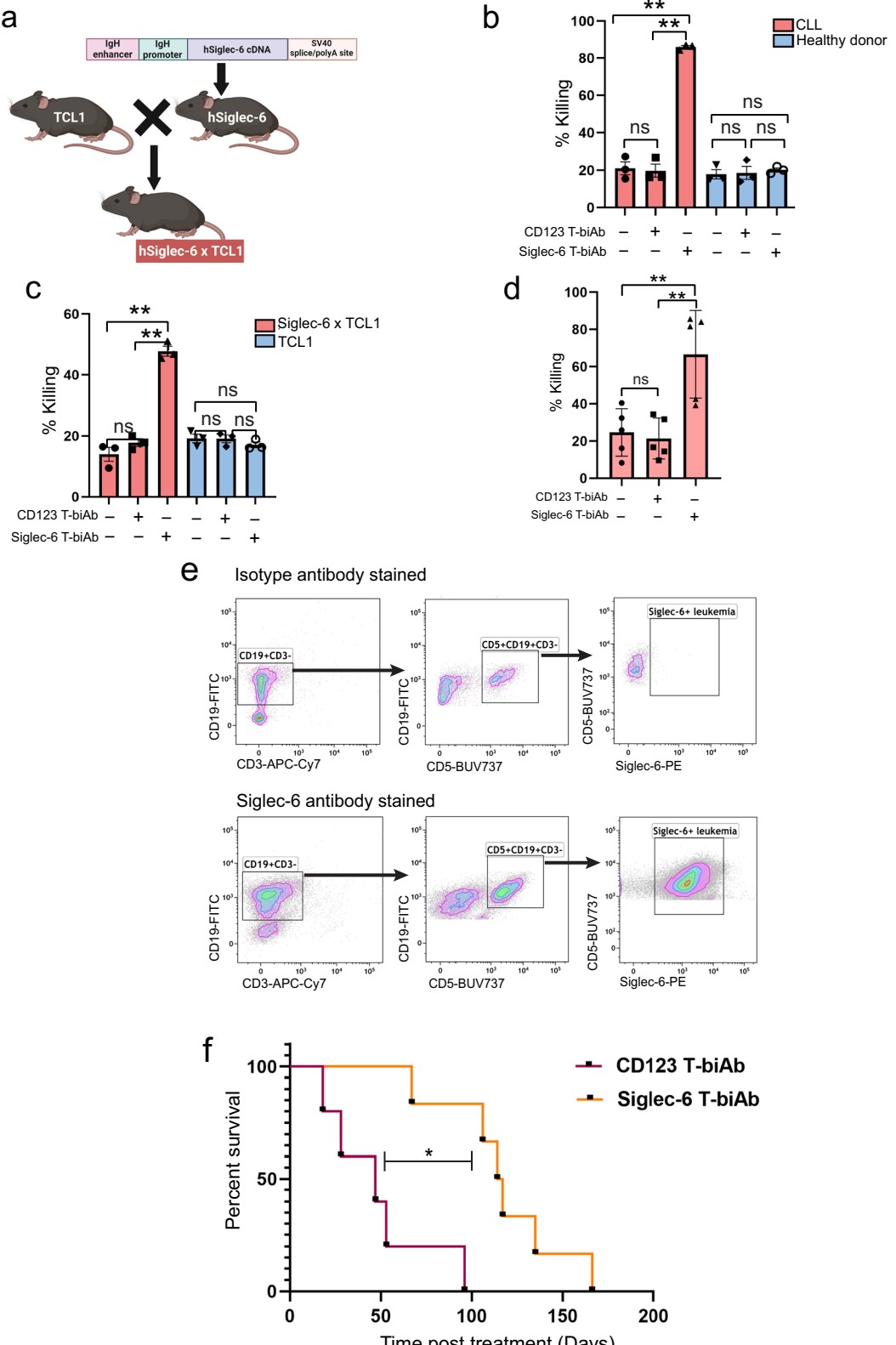

## In vivo bispecific antibody treatment study

We used the immunocompetent humanized CD3 (huCD3) mouse model to evaluate the efficacy of the Siglec-6 T-biAb. Six- to eight-week-old huCD3 mice were inoculated with $5 \times 10^6$ Siglec-6+ leukemic cells that were sorted from splenocytes of Siglec-6 × TCL1 mice via i.v. (tail vein) injection. Peripheral blood was collected via cheek bleed every week to measure the percent of circulating leukemic cells (CD5+ CD19+ CD3-). Once mice displayed >3% circulating leukemia, they were randomly assigned to either the non-targeting CD123 T-biAb group or the Siglec-6 T-biAb treatment group. Treatment was done once a week (0.1 mg/kg diluted in DPBS, 4 mice per group) via i.v. (tail vein) injection, until the mice met early removal criteria (ERC), defined by low body score, weight loss >20%, and hind limb paralysis. NSG mice were used to evaluate the

**Fig. 8 | Siglec-6 targeted T cell-recruiting antibody (T-biAb) selectively depletes leukemic B cells ex vivo and improves the overall survival of mice with hSiglec-6×TCL1 leukemia. a** Schema showing the generation of a hSiglec-6×TCL1 mouse model that expresses human Siglec-6 on leukemic B cells. **b, c** Target cells (B cells from 3 independent TCL1 mice, hSiglec-6×TCL1 mice, healthy donors, or CLL patients) were cocultured with activated huCD3 T cells from huCD3 mice (one 8–12 week male huCD3 mouse per target) and Siglec-6 or CD123 T-biAb. Cytotoxicity was measured by Annexin V /PI staining. The Siglec-6 T-biAb exhibited specific killing of Siglec-6 + CLL cells ($P = 0.0004$) and Siglec-6+ hSiglec-6 × TCL1 cells ($P = 0.002$) when compared to non-targeting CD123 T-biAb. **d** Treatment-naive CLL peripheral blood mononuclear cells (PBMCs) (5 patients) were cultured with T-biAbs for 7 days and analyzed by flow cytometry to determine specific killing of CLL cells by autologous T cells at endogenous effector (E) to target (T) (E:T) ratios. The Siglec-6 T-biAb exhibited specific killing of CLL cells

($P = 1.01 \times 10^{-21}$) when compared to CD123 T-biAb. **e** Flow data from a representative mouse showing the gating strategy to isolate Siglec-6 leukemic cells (Siglec-6+ CD5+ CD19+ CD3−) from hSiglec-6 × TCL1 mice. Siglec-6+ gate was determined based on staining with an isotype control Ab. **f** Siglec-6 T-biAb [0.1 mg/kg, once a week till end of study treatment (i.v.)] significantly prolongs survival of biologically independent leukemic mice [$n = 5$ (control CD123 T-biAb), $n = 6$ (Siglec-6 T-biAb), $P = 0.007$]. 11-month male and female TCL1 and hSiglec-6xTCL1 mice (C57BL/6 background) were used in (**c**); 11-month male hSiglec-6×TCL1 (C57BL/6 background) and 8–12 week male and female (equal ratio) huCD3 mice (B6.Cg-Tg (CD3E) 600Cpt/J) were used in (**f**). Graphs show the mean ± standard error of the mean. *$P < 0.05$; **$P < 0.005$ by two-sided mixed effect modeling (**b**–**d**) or log-rank test (**f**). ns not significant. Source data are provided in the source data file. **a** was created with BioRender.com.

T-biAb in a CLL patient-derived xenograft model[53]. $5 \times 10^7$ PBMCs from CLL patients were injected via tail vein, followed by confirmation of CLL engraftment on day 3 by peripheral blood flow cytometry. 6 mice were injected with cells from 3 CLL patients per group. Siglec-6 or CD123 T-biAbs (0.1 mg/kg diluted in DPBS) were then injected intraperitoneally. Flow cytometry was used to assess tumor burden in the peripheral blood on days 10 and 17, and in the spleen on day 17.

## Flow Cytometry

Flow cytometric experiments were performed using an LSRII Fortessa (BD Biosciences) flow cytometer. Flow cytometric data was analyzed using Kaluza Analysis 2.1 (Beckman Coulter, Indianapolis, IN) software. SYTOX blue dead cell stain or near IR (780) fluorescent reactive dye (Invitrogen) were used to exclude dead cells. Fluorochrome-labeled human (hu) or murine (m) mAbs were used to stain cells for analysis by flow cytometry. Staining was done in FACS buffer (PBS + 2% FBS). A full list of antibodies can be found in Supplementary Table 2. Human Siglec-6/CD327 PE-conjugated antibody (Clone #767329, R&D Systems) was used to stain Siglec-6 on patient samples, and IgG$_{2A}$ PE-conjugated Antibody (Clone #20102, R&D Systems) was used as control. Chemically biotinylated (BiotinTag Micro Biotinylation Kit; Sigma-Aldrich) JML-1 human IgG1[7] followed by staining with streptavidin PE (Thermo Fisher) was used for all other Siglec-6 surface staining experiments, and biotinylated human IgG1 isotype mAb was used as control. Absolute cell concentrations were obtained by quantitative flow cytometry using CountBright absolute counting beads (Invitrogen).

## siRNA knockdown

Transient transfection of MEC1-002 cells was performed using the Lonza Cell Line Nucleofector™ Kit V (# VCA-1003) in the Nucleofector™ I/II/2b Device (Lonza) according to the manufacturer's specifications. Briefly, $1 \times 10^6$ cells were resuspended in 100 µl of nucleofector solution and mixed with the relevant siRNA. Nucleofection was performed using program M-13. Cells were rapidly transferred to a preheated complete medium (RPMI 1640 + 10% FBS) and incubated for 48 h at 37 °C. For Siglec-6 knockdown, 500 nM Siglec-6-specific SMARTpool siRNAs from Dharmacon (#E-004226-00-0010) were used. Target sequences are as follows: GGAUGAAAUACGGUUAUAC (#A-004226-4), CGUAUAGUUUCAGAUGUUA (#A-004226-15), CCCAUGACCUAAUUUAAAU (#A-004226-16), and GCCAUAACCUUGAUUGGAG (#A-004226-17). For non-targeting control, 500 nM of Accell Green Non-Targeting siRNA (#D-001950-01-05) with sequence UGGUUUACAUGUCGACUAA was used. For SHP-2 knockdown, 150 nM PTPN11 siRNA (AM51331) (sense strand sequence: CCAUGUUAUGAUUCCUGUtt, antisense strand sequence: ACAGCGAAUCAUAACAUGGgt) and related negative control (Silencer® Negative Control #1 siRNA) from Thermo Fisher was used.

## FACS sorting

Cell sorting was performed using a BD Bioscience BD FACSAria™ Fusion flow cytometer with a blue, red, yellow-green, violet, and UV laser (BD Biosciences). Splenocytes from Siglec-6 x TCL1 mice were selected and sorted for with the following antibody cocktail: mCD19-FITC (6D5) from Biolegend; mCD5-BUV737 (53-7.3) and mCD3-APC-Cy7 (145-2C11) from BD Biosciences; SYTOX blue dead cell stain (Invitrogen); and biotinylated JML-1 human IgG1 followed by staining with streptavidin PE (Thermo Fisher) for Siglec-6 staining. Siglec-6+ leukemic cells were identified as live, CD19+, CD5+, CD3−, and Siglec-6+. Briefly, cells were stained with an antibody cocktail on ice for 1 h before being washed once and resuspended in RPMI media. Cells were sorted through an 85 µm nozzle at a sheath pressure of 25 psi and a drop drive frequency of 60–70 kHz. 4-way purity sorting was used. Sorted purity achieved for all samples was >98.0%.

## In vitro migration and adhesion assay

12-h migration assays were conducted using 6 µm pore size 24-well transwell plates (Corning). $0.2 \times 10^6$ BMSCs were plated in the bottom chambers and allowed to attach overnight. MEC1-002 or primary B cells in ($1 \times 10^6$ in 100 µl serum free media) were added to the upper chamber and allowed to migrate towards BMSCs for 12 h at 37 °C. Transwell inserts were then removed, and migrated cells were counted using CountBright absolute counting beads (Invitrogen) using an LSRII Fortessa (BD Biosciences) flow cytometer. Fold migration was calculated vs baseline migration without stromal cells. In graphs where percent migration is displayed, percent migration was calculated by normalizing to the total number of cells seeded. For adhesion assay, $0.2 \times 10^6$ BMSCs were plated in 24-well plates and allowed to attach overnight. The following day, $1 \times 10^6$ MEC1-002 or primary B cells in RPMI media were seeded onto the BMSCs and allowed to adhere for 12 h at 37 °C. Unattached cells were then washed twice and the cells left behind were trypsinized and counted using CountBright absolute counting beads (Invitrogen) using an LSRII Fortessa (BD Biosciences) flow cytometer. Fold attachment was calculated vs baseline attachment without stromal cells. In graphs where the percent attached is displayed, the percent attached was calculated by normalizing to a total number of cells seeded. In both assays, cells were blocked with either JML-1 anti-Siglec-6 antibody or human IgG1 isotype control; or treated with ibrutinib (1 µM) or DMSO control for 1 h before seeding into the transwells/onto BMSCs. Human CD19 PE (6D5) was used to identify and count only CD19+ B cells.

## Confocal imaging

Cells were centrifugally concentrated on microscope slides using a Cytospin3 centrifuge (Thermo, Asheville, NC). Cells were then fixed in PBS/2% paraformaldehyde and permeabilized with 0.2% Tween 20. Slides were incubated in blocking solution (1% bovine serum albumin in PBS + 0.1% Tween 20) and stained for Siglec-6 (Sigma-Aldrich, #

a

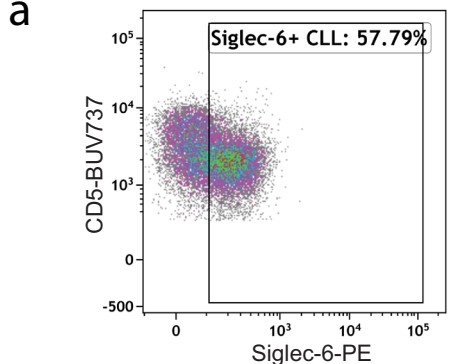
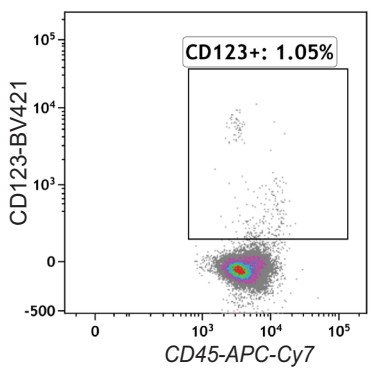

b

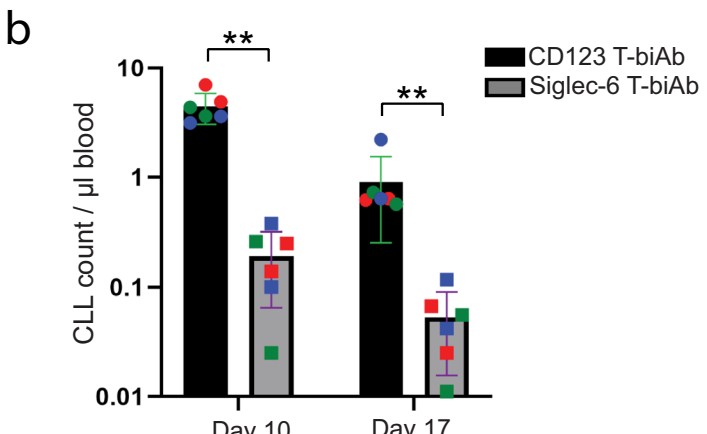

c

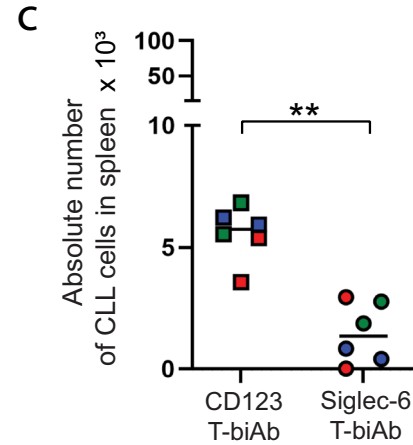

d

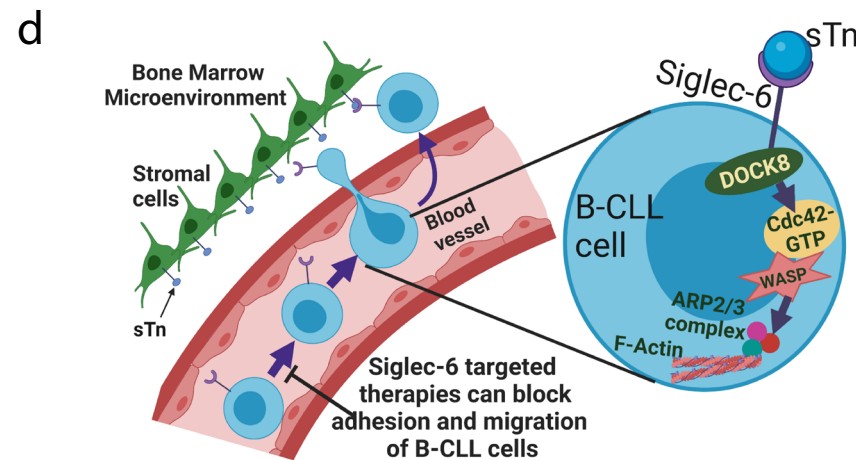

**Fig. 9 | Siglec-6 targeted T-biAb eliminates chronic lymphocytic leukemia (CLL) cells in patient-derived xenografts. a** Representative flow cytometry plots showing Siglec-6 expression on CLL cells, and limited CD123 expression on patient peripheral blood mononuclear cells (PBMCs) from 3 patients. **b** CLL cell count in the peripheral blood at days 10 and 17. **c** CLL cell count in the spleen at experimental day 17. For (**b, c**), each dot represents 1 animal; the color code denotes patient source of xenografted cells which correlates with colors. 3 independent CLL patients and six 8-week male NOD.Cg-Prkdc[scid] Il2rg[tm1Wjl]/SzJ (NSG) mice were used. **d** Schema showing the mechanism of sTn-mediated Siglec-6 dependent migration of Siglec-6+ cells. Targeting Siglec-6 and blocking migration to the TME thus provides a novel therapeutic opportunity to treat CLL patients who develop resistance to existing therapies. Graphs show the mean ± standard error of the mean. *$P < 0.05$; **$P < 0.005$ by two-sided mixed effect modeling (**b, c**). ns not significant. Source data are provided in the source data file. **d** was created with BioRender.com.

HPA009084) or DOCK8 (Origene, #TA506484) by incubating with the primary antibodies overnight at 4 °C, followed by incubation with secondary antibodies, goat anti-rabbit IgG Alexa Fluor 488 (Invitrogen) or goat anti-mouse IgG1 Alexa Fluor 594 (Invitrogen), respectively. Texas Red-X phalloidin (Invitrogen) was used to stain F-actin during the secondary antibody incubation step. Nuclei were stained blue with DAPI (Vector Laboratories). Olympus Fluoview 1000 Laser Scanning Confocal microscope at the Ohio State University Campus Microscopy and Imaging Facility was used for capturing images. For Fig. 6c and supplementary Fig. 7a, images were taken using a ×60 objective, and z-stacking was performed at pixel resolution 512 × 512. For Fig. 1a, and supplementary Fig. 4b, images were captured using a ×20 objective at pixel resolution 512 × 512. Equipment specifications are as follows: (a) Upright Microscope: Olympus BX61F/BX62F; (b) Detectors: 4 filter-based fluorescence PMT detectors and one transmitted, DIC detector; (c) Available Objectives: UPLSAPO 10×, N.A. 0.4 LUCPLFLN 20×, N.A. 0.45 UPLSAPO 20× Oil, N.A. 0.85 UPLFLN 40× Oil, N.A. 1.3 PLAPONSC 60× Oil, N.A. 1.4; (d) Image info: OIB or OIF Olympus format, 12 bit grayscale 24 bit color, JPEG/BMP/TIFF/AVI image formats also available, and pixel resolution formats range from 64 × 64 up to 4096 × 4096.

## Co-culture immunofluorescent microscopy

For live co-culture adhesion assays, BMSCs and CLL/healthy donor B cells were labeled with BioTracker 490 Green or BioTracker 655 Red cytoplasmic membrane dyes respectively, according to the manufacturer's instructions (MilliporeSigma). Co-culture immuno-fluorescence images were captured using the ECHO Revolve R4 microscope, at 10× magnification.

## Cell Fractionation

$10 \times 10^6$ MEC1-002 (WT, Siglec-6 KO or DOCK8 KO), B-CLL or normal B cells were plated in 12 well plates, treated with 6 µg sTn for 1 h. Membrane, nuclear, and cytoplasmic fractions were then prepared according to the manufacturer's protocol (Thermo Fisher #78840).

## Cdc42 activation assay

$10 \times 10^6$ MEC1-002 (WT, Siglec-6 KO or DOCK8 KO) cells were plated in 12 well plates, treated with 6 µg sTn for 1 min or blocked with 10 µg anti-Siglec-6 mAb for 30 min followed by sTn treatment for 1 min. Cdc42 activation assay was carried out according to the manufacturer's protocol (Abcam, ab211163)

## Siglec-6-sTn interaction studies

For ELISA-based assay, Pierce™ Protein A Coated Plates (Thermo Scientific) were coated with 2 µg/ml Siglec-6 Fc fusion protein or IgG Fc (Acro Biosystems) in 50 mM carbonate bicarbonate buffer (pH 9.5), overnight at 4 °C. After blocking for 1 h at room temperature with PBS/2%BSA, biotinylated sTn (sTn-B) (GlycoNZ, #0058BP) was added at different concentrations. Binding was measured using tetramethyl benzidine substrate (R&D Systems) according to the manufacturer's instructions. For cell line-based assay, the siglec negative DT-40 cell line was transfected either with 1 µg CMV-Siglec-6 construct or CMV empty vector[38] using the Amaxa Nucleofector system. 24 h post-transfection, cells were either blocked with isotype mAb or Siglec-6 blocking mAb, followed by sequential probing with sTn-B and streptavidin PE. Median fluorescence intensity as a readout of binding was measured via flow cytometry analysis. To test the binding of Siglec-6 to sTn on CLL-BMSCs, CLL-BMSCs were blocked with anti-Sialyl Tn antibody [STn 219] (Abcam) or IgG1 isotype control for 30 min at 4 °C. Cells were also treated with 0.6 U sialidase (MilliporeSigma) for 30 min at 37 °C[65]. Following this, cells were washed, and Siglec-6 Fc fusion protein was added for 30 min at 4 °C. After this, cells were washed and stained with biotinylated JML-1 for 1 h, after which cells were washed again and stained with PE-Cy7 Streptavidin (BD Biosciences) for

30 min. Flow cytometry was used to quantify Siglec-6 binding to CLL-BMSC based on fluorescence in the PE-Cy7 channel.

## Western blotting

For all western blotting experiments, cells were lysed with RIPA buffer (CellSignaling Technology, #9806) supplemented with protease and phosphatase inhibitors. Lysate concentration was determined using Pierce™ BCA Protein Assay Kit (ThermoFiser Scientific, #23225) and proteins were analyzed via western blotting. A full list of antibodies can be found in the inventory of supporting information files. Images were captured using SuperSignal™ West Pico PLUS Chemiluminescent Substrate (#34577).

## Co-immunoprecipitation (Co-IP)

Co-IP experiments were performed according to the manufacturer's instructions (Dynabeads™ Protein G Immunoprecipitation Kit, Thermo Fisher). Anti-human Siglec-6 antibody (clone 2G6, LifeSpan Biosciences) or mouse IgG2b isotype control antibody was used to pull-down Siglec-6 protein, and followed by immunoblotting with DOCK8 antibody (EPR1251, Abcam). Rabbit Siglec-6 (polyclonal, Abcam) was used to confirm Siglec-6 pull-down via immunoblotting. For co-IP experiments with BMSCs, Siglec-6 Fc fusion protein or IgG Fc (Acro Biosystems) was conjugated to Protein A agarose beads (MilliporeSigma) to test the interaction of Siglec-6 with sTn+ proteins. Pull-down of sTn+ proteins was confirmed using anti-human sialyl Tn antibody (B35.1, LifeSpan Biosciences). Multiple bands were observed in both western blotting and IP experiments with sTn corresponding to a band detected at ~150 kDa according to company product specifications. Anti- Siglec-6 antibody (polyclonal, Abcam) was used to confirm the conjugation of Siglec-6 to Protein A agarose beads.

## B cell homing assay

NOD.Cg-Prkdc$^{scid}$ Il2rg$^{tm1Wjl}$/SzJ (NSG) mice were obtained from Jackson Laboratory (005557). B-CLL or MEC1-002 cells were injected into the tail vein of 4- to 8-week-old NSG mice ($10 \times 10^6$ cells per mouse). For blocking experiments, B-CLL or MEC1-002 cells were incubated 1 h before injection with JML-1 mAb or human IgG1 isotype control antibody. 24 h after injection, mice were sacrificed, and bone marrow and spleen were collected. Human B cells from the different organs were detected by flow cytometry using human-specific anti-CD45 antibody and an absolute number of migrated cells was counted using CountBright absolute counting beads (Invitrogen).

## T cell expansion

Humanized mouse T cells (huCD3 T cells) were expanded from humanized CD3ε mouse which expresses the human epitope of the CD3ε chain[51,52] by seeding at 1 million cells in 200 µL IMDM media with concanavalin A (Invitrogen) in the presence of 100 U/mL IL-2 (R & D systems) and 2-mercaptoethanol over a 7-day culture.

## Ex vivo cytotoxicity assay

B cells were isolated from the spleens of TCL1 or hSiglec-6 × TCL1 mice using the EasySep™ Mouse B Cell Isolation Kit (StemCell Technologies). RosetteSep enrichment kit (#15064, StemCell Technologies) was used to isolate B cells from CLL and healthy donor whole blood samples. Target B cells were labeled with CellTrace violet (Invitrogen) and cultured with concanavalin A activated effector huCD3 T cells, along with the CD123 control T-biAb or Siglec-6 T-biAb (6 nM) for 5 h. Effector: target ratio used was 5:1. CD123 T-biAb was obtained from Xencor (Xmab14045), and Siglec-6 T-biAb in scFv-Fc format (silenced Fc) based on mAb JML-1 was recently described[38]. Cells were then stained with Annexin V and propidium iodide (PI) (Leinco Technologies) to calculate % target dead cells by flow cytometry. To test the Siglec-6 T-biAb against treatment-naïve CLL patients using autologous T cells, PBMCs from 5 CLL patients were isolated using Ficoll density-

gradient centrifugation and were cultured with either Siglec-6 targeting T-biAb (6 nM) or CD123 T-biAb (6 nM) for 7 days. The percentage of CLL cells (CD5+ CD19+ CD3−) that were killed was measured using annexin V/PI staining and flow cytometry analysis. % Killing = (100 − % Annexin V and PI negative cells).

## Statistics and reproducibility
All proteomics samples were analyzed using Mascot (Matrix Science) and a two-tailed student $t$-test was performed. Proteins with a $P$-value less than 0.05 (or 5%) were considered to be significantly changed. Proteins with a fold change >2 or <0.5 were considered as up- and down-regulation respectively. All other statistical analyses were performed with SAS 9.4 (SAS Inc.) at the OSU Center for Biostatistics. Statistical analyses were completed using mixed effects models (two-sided) and paired two-tailed $t$-tests. For in vivo survival studies, log-rank tests were used to compare survival probabilities across different treatment groups. No statistical method was used to predetermine sample size, no data were excluded from the analyses, and investigators were not blinded to allocation during experiments and outcome assessment. $P < 0.05$ was considered significant. ImageJ was used for quantification of immunoblots and confocal microscopy images.

## Reporting summary
Further information on research design is available in the Nature Portfolio Reporting Summary linked to this article.

## Data availability
The authors declare that the data supporting the findings of this study are available within the paper and its supplementary information files. All the raw data generated in this study are provided in the Source Data file. The mass spectrometry data was generated for this study and has been deposited in the ProteomeXchange database (Project accession: PXD039892). The complete list of proteins identified in the mass spectrometry experiment is provided in supplementary data 1. Other relevant data that further support the findings of this study are available from the corresponding author upon request. Source data are provided with this paper.

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

## Acknowledgements

The authors thank the patients who contributed to these studies; Christopher Mannings of the OSU Comprehensive Cancer Center Leukemia Tissue Bank Shared Resource; John Byrd, University of Cincinnati, for his intellectual contribution; Elizabeth Perry and Kevan Zapolnik for their help with mouse experiments; and Brian Kemmenoe, Campus Microscopy & Imaging Facility (CMIF) at OSU, for his help with confocal microscopy analysis. We acknowledge resources from the CMIF, and the OSU Comprehensive Cancer Center (OSUCCC) Microscopy Shared Resource (MSR) at OSU with NIH S10 OD025008 and NIH NIC P30CA016058. The Fusion Orbitrap instrument was supported by NIH Award Number Grant S10 OD018056. This work was supported by NIH, National Cancer Institute (NCI) grant R21 CA229961-01 (MPIs: Rader/Muthusamy), and Pelotonia Idea grant (Muthusamy). J.N. was funded by a Pelotonia Graduate Fellowship. This study received funding support from the following: National Cancer Institute (NCI), National Institutes of Health (NIH), through grant R21 CA229961-01 awarded to N.M. and C.R.; Pelotonia Idea Grant awarded to N.M.; Macbea Foudation award to NM and Graduate Pelotonia fellowship awarded to J.N.

## Author contributions

J.N. planned and designed the experiments, analyzed the data, and wrote the drafts of the article; R.T. assisted with experiments; X.M. performed the statistical analysis; C.M. and M.P. performed the FACS sorting experiments; L.Z. prepared samples for mass spectrometry analysis and generated the raw data; M.L. provided huCD3 mice and necessary expertise with the bispecific antibody related experiments; C.R. and M.G.C. generated the human JML-1 monoclonal antibody and Siglec-6 T-biAb, and provided the MEC1-002 cell line; and N.M. conceived the idea, designed the experiments, supervised the study, sought funding, reviewed drafts, and approved the final version for submission.

## Competing interests

The authors C.R. and M.G.C. are inventors of two patent applications for recombinant human antibodies targeting Siglec-6 and their applications, listed below: (i) U.S. Patent 8,877,199 for B Cell Surface Reactive Antibodies. Assignee: U.S. Department of Health and Human Services (Washington, DC, USA). (ii) PCT/US2022/079656 for Siglec-6 Antibodies, Derivative Compounds and Related Uses. Assignee: University of Florida Research Foundation, Inc. (Gainesville, FL, USA). The other authors do not declare any competing interests.
