## [Peer Review File · Nature Communications]

REVIEWER COMMENTS

Reviewer #1 - CLL migration, therapy (Remarks to the Author):

This manuscript investigated the putative role of Siglec-6 in chronic lymphocytic leukaemia (CLL). The authors identify a novel role for Siglec-6 in the migration and adhesion of CLL cells. They went on to investigate the therapeutic potential of using a Siglec-6-targeted bi-specific antibody. The manuscript is generally well-written and the experimental approaches employed appear to be rigorous and well-performed. However, a number of issues remain unresolved, which limits the manuscript's potential impact:

1. The authors claim that normal B cells do not express Siglec-6. However, this is contradicted by Koralovsky et al (ref 24) and the Protein atlas data (proteinatlas.org). This discrepancy may be due to the fact that memory B cells appear to express more Siglec-6 than naive B cells. The authors should comment on this, not least because it has implications for the elimination of memory B cells in the context of Siglec-6-targeting therapy.
2. Many of the experiments are based around the use of the MEC1-002 cell line - a sub-line derived from MEC1 cells. The MEC1 cells used in their study (Fig S1b) do not show the same pattern Siglec-6 expression as the MEC1 cells from which the MEC1-002 cells were derived (Chang et al. Cancer Immunol. Res. (2019)). This reviewer would have preferred at least some of the comparative experiments to be performed on FACS sorted Siglec-6 high and low subsets derived from the parental line.
3. Given the stated role of Siglec-6 in modulating CLL cell migration and adhesion, it would have been important to examine the expression of Siglec-6 in prognostic subsets of CLL with different migratory and adhesion potential i.e. CD49dhi and CD49dlo. It would also have been very interesting to know whether CXCR4lo/CD5hi CLL cells - so-called "recent emergent" cells from the lymphoid tissues, showed different Siglec-6 expression when compared with CXCR4hi/CD5lo CLL cells.
4. Figure 3 shows an incredibly high level of MEC1-002 cell migration (>70%) - this level of migration would not be observed in primary CLL cell samples over a 12h assay period. At the least, the authors should have compared MEC1-002 cell migration with primary CLL samples +/- Siglec-6 blockade with JML-1.
5. Although the in vivo targeting data, using a bi-specific antibody, look promising, this reviewer would have preferred to see some in vitro data using autologous T cells from CLL patients. The humanised CD3 transgenic mouse T cells are not an appropriate proxy for T cells derived from patients. This is important because it is very well established that CLL patients' T cells possess a number of defects, which can limit their cytotoxic effect, even in the context of a forced immune synapse (as caused by a bi-specific antibody).
6. As above, it may have been more clinically relevant to conduct the in vivo experiments in the context of a PDX model of CLL, created using PBMC from patients. Such a model would contain representative numbers of patient-derived tumour cells and T cells.

7. Given the class effect of Bruton's Tyrosine Kinase inhibitors, like ibrutinib, it would have been interesting for the authors to consider whether exposure to ibrutinib alters Siglec-6 cell surface expression and whether the combination of JML-1 and ibrutinib is synergistic in the context of inhibiting CLL cell adhesion and/or migration.

Reviewer #2 - Glycosylation, siglec, immune cells (Remarks to the Author):

In the manuscript by Nunes et al. the authors investigate the CLL-specific expression of Siglec-6. They claim that Siglec-6 can bind sTn to facilitate migration and homing to the bone marrow. By stimulating Siglec-6+ cells they elucidate a role for Siglec-6 in DOCK8 and cdc42 activation, WASP recruitment and actin polymerization. Finally, using a Siglec-6/CD3 bispecific antibody the researchers show the potential of this antibody for the treatment of CLL.

In my opinion the authors make a compelling case that Siglec-6 is specifically expressed in the CLL cells. Moreover, their results convincingly show that Siglec-6 mediates migration and binding of CLL to BMSCs. Nevertheless, my main concern centers around the sTn-specificity. Yes, Siglec-6 can bind sTn, a finding which was already published before. Indeed, also the BMSCs from CLL patients express very low levels of sTn, yet the authors fail to prove that it is indeed sTn that mediates the binding to the BMSCs. Additional ligands of Siglec-6 have been identified before and some of them are even carbohydrate-independent. Therefore, the authors should show that IPs with Siglec-6-Fc from BMSCs lysates yields sTn positive proteins. Moreover, sTn blocking studies should be conducted. The authors could use sialidase treatment to show that the binding is sialic acid mediated and preferentially even knock-down/knockout of ST6GalNAC1 to abolish sTn expression overall.

In addition to this I have a few other questions for the authors:

- Is the Siglec-6 mediated triggering of the DOCK8/WASP axis dependent on SHP-2, a known downstream signaling component of the Siglec-6 signaling pathway? Does a KO of SHP-2 also abolish the downstream actin polymerization?

- In Fig.3B the authors do not observe sTn-induced changes in DOCK8 expression in B-CLL cells. Have the authors considered that Siglec-6 might be (partially) masked in these cells? Masking of Siglecs by cis-expressed sialic acids is commonly observed in many cell types. Are the B-CLL cells sialylated and if so, does sialidase treatment now allow Siglec-6 to alter DOCK8 expression upon sTn triggering?

- Does Fig. 6B show technical replicates or does the figure show the combined data of 3 individual donors? This was unclear to me.

- In humans Siglec-6 is also highly expressed by mast cells. The Siglec-6/CD3 bispecific antibody would in vivo also target these cells. Do the authors anticipate specific mast cell-associated side effects of their treatment and if so to what degree? Could the authors comment on this in their discussion?

Reviewer #3 - Blood cancer therapy, proteomics, mouse models (Remarks to the Author):

The authors study the expression, molecular network and therapeutic targeting of SIGLEC-6 in CLL models and primary cells. The target has previously been linked to therapeutic targeting, so the novelty and relevance would need to emerge from the molecular understanding of its function. The authors demonstrate a role in migration/homing and work-up some aspects of the molecular network including DOCK8 and cdc42 mainly in MEC1/2 cell lines.

To me the study is of interest, but the relevance is decreased as it is not clear to me what the pathophysiological relevance of the pathway is. The mechanisms controlling migration are manyfold. Do the authors suggest this mechanism is relevant for marrow/LN homing and how is this particular pathway related to many other mechanisms of homing/migration.

The therapeutic potential is of interest, but not clearly related to the main focus of the study.

Siglec-6: A novel therapeutic target of cell migration and adhesion in Chronic Lymphocytic Leukemia"

Reviewer comments are highlighted in italics. Changes to the manuscript text are highlighted in yellow and the corresponding page, line and figure numbers are provided below and highlighted in bold.

Reviewer 1:

1. The authors claim that normal B cells do not express Siglec-6. However, this is contradicted by Koralovsky et al (ref 24) and the Protein atlas data (proteinatlas.org). This discrepancy may be due to the fact that memory B cells appear to express more Siglec-6 than naive B cells. The authors should comment on this, not least because it has implications for the elimination of memory B cells in the context of Siglec-6-targeting therapy.

Response: We thank the reviewer for bringing this to our attention. We apologize for the oversight in the previous description since we meant to say that Siglec-6 is overexpressed on B cells from CLL patients when compared to normal B cells. Indeed, flow cytometry analysis of CD19+CD5+ B cells from CLL patients revealed varying levels of surface expression of Siglec-6 ranging from 2.45-749.8 median fluorescence intensity (MFI) (n=31, mean MFI 103.13). In contrast, CD19+ B cells from healthy donors revealed minimal Siglec-6 expression ranging from 1.11-32.84 MFI (n=10, mean MFI: 11.9). We have modified the description accordingly (**Page 4, Line 90**). Additionally, we have included information about Siglec-6 expression on memory B cells in the introduction section (**Page 3, Line 53-57**) and have discussed the implications of targeting Siglec-6+ memory B cells in the discussion section (**Page 19, Line 403-412**).

2. Many of the experiments are based around the use of the MEC1-002 cell line - a sub-line derived from MEC1 cells. The MEC1 cells used in their study (Fig S1b) do not show the same pattern Siglec-6 expression as the MEC1 cells from which the MEC1-002 cells were derived (Chang et al. Cancer Immunol. Res. (2019). This reviewer would have preferred at least some of the comparative experiments to be performed on FACS sorted Siglec-6 high and low subsets derived from the parental line.

Response: We thank the reviewer for raising this important point. The MEC1-002 that we have used in our study was kindly provided by Dr. Christoph Rader, the corresponding author of the original paper that characterized the MEC1-002 cell line (Chang *et al.*, Cancer Immunol. Res. 2019). The MEC1 cells that are depicted in our manuscript were not used in any of our experiments, but was merely shown to compare and demonstrate high Siglec-6 levels in the MEC1-002 cells. The Siglec-6 peak in the MEC1 cells used by Chang *et al* to generate the MEC1-002 cells was not seen in the MEC1 cells that we tested from our lab, which could be attributed to differences in passage number. Nevertheless, for all our subsequent experiments to study the biological role of Siglec-6, we knocked out Siglec-6 in the original MEC1-002 cell line to generate a MEC1-002 Siglec-6 knock-out cell line. We feel that the comparative experiments that we have

performed between the original MEC1-002 cells and the MEC1-002 Siglec-6 knock-out cells are sufficient to demonstrate the biological role of Siglec-6.

3. *Given the stated role of Siglec-6 in modulating CLL cell migration and adhesion, it would have been important to examine the expression of Siglec-6 in prognostic subsets of CLL with different migratory and adhesion potential i.e. CD49dhi and CD49dlo. It would also have been very interesting to know whether CXCR4lo/CD5hi CLL cells - so-called "recent emergent" cells from the lymphoid tissues, showed different Siglec-6 expression when compared with CXCR4hi/CD5lo CLL cells.*

Response: We thank the reviewer for this interesting question. Our evaluation of Siglec-6 expression on CD49dhi, CD49dlo, CXCR4lo/CD5hi and CXCR4hi/CD5lo CLL cells among 6 CLL patient samples did not reveal any significant differences in Siglec-6 expression in these subsets (**Page 9, Line 183-193**) and **Supplementary Fig. 2a and 2b**).

4. *Figure 3 shows an incredibly high level of MEC1-002 cell migration (>70%) - this level of migration would not be observed in primary CLL cell samples over a 12h assay period. At the least, the authors should have compared MEC1-002 cell migration with primary CLL samples +/- Siglec-6 blockade with JML-1.*

Response: We thank the reviewer for pointing this out. We have conducted the *in vitro* migration assay as was suggested by the reviewer, by comparing MEC1-002 migration with primary samples +/- Siglec-6 blockade (**Supplementary Fig. 1e**). MEC1-002 cells were found to migrate less when compared to primary CLL cells. For all the *in vitro* migration and adhesion assays, we have reported extent of migration and adhesion in terms of fold change, as described previously¹.

5. *Although the in vivo targeting data, using a bi-specific antibody, look promising, this reviewer would have preferred to see some in vitro data using autologous T cells from CLL patients. The humanised CD3 transgenic mouse T cells are not an appropriate proxy for T cells derived from patients. This is important because it is very well established that CLL patients' T cells possess a number of defects, which can limit their cytotoxic effect, even in the context of a forced immune synapse (as caused a bi-specific antibody).*

Response: We agree with the reviewer that this is a crucial aspect in the study of Siglec-6 therapeutic targeting. As suggested, we have tested the huSiglec-6 x huCD3 bispecific antibody against PBMCs from treatment-naïve CLL patients (**Page 14 and 15, Line 313-317, Fig. 7d**).

6. *As above, it may have been more clinically relevant to conduct the in vivo experiments in the context of a PDX model of CLL, created using PBMC from patients. Such a model would contain representative numbers of patient-derived tumour cells and T cells.*

Response: In continuation with the above experiment, we agree with the reviewer that testing the Siglec-6 bispecific antibody in a CLL PDX model would be more clinically relevant and have carried out this experiment as suggested (**Page 15 and 16, Line 331-342, Fig. 8c-e**).

7. Given the class effect of Bruton's Tyrosine Kinase inhibitors, like ibrutinib, it would have been interesting for the authors to consider whether exposure to ibrutinib alters Siglec-6 cell surface expression and whether the combination of JML-1 and ibrutinib is synergistic in the context of inhibiting CLL cell adhesion and/or migration.

Response: We thank the reviewer for this suggestion. CLL patients undergoing ibrutinib treatment do not show altered surface Siglec-6 expression when compared to pre-ibrutinib treatment, as reported by Cyr *et al*². We did not observe any synergistic effect with the combination of JML-1 and ibrutinib on CLL cell migration and adhesion (**Page 9, Line 194-202, Fig. 3f and 3g**).

Reviewer 2:

1) In my opinion the authors make a compelling case that Siglec-6 is specifically expressed in the CLL cells. Moreover, their results convincingly show that Siglec-6 mediates migration and binding of CLL to BMSCs. Nevertheless, my main concern centers around the sTn-specificity. Yes, Siglec-6 can bind sTn, a finding which was already published before. Indeed, also the BMSCs from CLL patients express very low levels of sTn, yet the authors fail to proof that it is indeed sTn that mediates the binding to the BMSCs. Additional ligands of Siglec-6 have been identified before and some of them are even carbohydrate-independent. Therefore, the authors should show that IPs with Siglec-6-Fc from BMSCs lysates yields sTn positive proteins. Moreover, sTn blocking studies should be conducted. The authors could use sialidase treatment to show that the binding is sialic acid mediated and preferentially even knock-down/knockout of ST6GalNAC1 to abolish sTn expression overall.

Response: We thank the reviewer for raising these critical comments that enabled us improve the manuscript. As suggested by the reviewer we conducted IP experiments with Siglec-6-Fc from CLL and normal donor BMSCs lysates. These results are included in the revised version (**Page 5, Line 110-111, Fig. 1d; Page 6, Line 122-128, Fig. 1g**). To further evaluate Siglec-6 binding to sTn via blocking studies, we conducted flow cytometry experiments. sTn antibody was used to block sTn and PE-conjugated Siglec-6 antibody was used to detect Siglec-6-Fc binding to sTn. Sialidase treatment was also done to test binding of Siglec-6 Fc (**Page 6, Line 128-135, Fig. 1h**). Technical difficulties precluded us from carrying out knock-down of ST6GalNAC1 in primary stromal cells.

2) Is the Siglec-6 mediated triggering of the DOCK8/WASP axis dependent on SHP-2, a known downstream signaling component of the Siglec-6 signaling pathway? Does a KO of SHP-2 also abolish the downstream actin polymerization?

Response: We thank the reviewer for bringing up this interesting point. We conducted SHP-2 knock-down experiments as suggested, and we did not observe compromised Cdc42 activation, WASP recruitment and actin polymerization, indicating that Siglec-6 is signaling through DOCK8 through a pathway that is independent of SHP-2 (**Page 12, Line 264-270, Supplementary Fig. 5**).

3) *In Fig.3B the authors do not observe sTn-induced changes in DOCK8 expression in B-CLL cells. Have the authors considered that Siglec-6 might be (partially) masked in these cells? Masking of Siglecs by cis-expressed sialic acids is commonly observed in many cell types. Are the B-CLL cells sialylated and if so, does sialidase treatment now allow Siglec-6 to alter DOCK8 expression upon sTn triggering?*

Response: We thank the reviewer for pointing this out. While sTn-induced changes in DOCK8 expression in B-CLL cells was not statistically significant, we observed an average of 2-fold increase in DOCK8 expression. The lack of statistical significance could be due to high variation due to patient heterogeneity. We repeated the experiment with two more CLL patient and healthy donor samples (n=5 in total), and the results show a statistically significant increase in DOCK8 expression in the membrane fraction of CLL cells upon sTn stimulation (**Page 11, Line 228-232, Fig. 4b**).

4) *Does Fig. 6B show technical replicates or does the figure show the combined data of 3 individual donors? This was unclear to me.*

Response: We apologize for the lack of clarity. The figure shows data from 3 individual donors. We have provided details in the figure legend. Note: **Fig. 6b** is now **Fig. 7b**.

5) *In humans Siglec-6 is also highly expressed by mast cells. The Siglec-6/CD3 bispecific antibody would in vivo also target these cells. Do the authors anticipate specific mast cell-associated side effects of their treatment and if so to what degree? Could the authors comment on this in their discussion?*

Response: We thank the reviewer for raising this important question. Mast cell associated side effects are likely to be expected with Siglec-6 targeted therapeutics. We have included comments on this in the discussion section (**Page 19, Line 403-412**).

Reviewer 3:

The authors study the expression, molecular network and therapeutic targeting of SIGLEC-6 in CLL models and primary cells. The target has previously been linked to therapeutic targeting, so the novelty and relevance would need to emerge from the molecular understanding of its function. The authors demonstrate a role in migration/homing and work-up some aspects of the molecular network including DOCK8 and cdc42 mainly in MEC1/2 cell lines. To me the study is of interest, but the relevance is decreased as it is not clear to me what the pathophysiological relevance of the pathway is. The mechanisms controlling migration are

manyfold. Do the authors suggest this mechanism is relevant for marrow/LN homing and how is this particular pathway related to many other mechanisms of homing/migration. The therapeutic potential is of interest, but not clearly related to the main focus of the study.

Response: We thank the reviewer for this extremely relevant point. We have included details in the discussion section to talk about pathophysiological relevance of this pathway, and how it relates to other mechanisms of homing/migration (**Page 17 and 18, Line 362-370 and 378-384**).

References:

- 1 Dubovsky, J. A. *et al.* Lymphocyte cytosolic protein 1 is a chronic lymphocytic leukemia membrane-associated antigen critical to niche homing. *Blood* **122**, 3308-3316 (2013). <https://doi.org/10.1182/blood-2013-05-504597>
- 2 Cyr, M. G. *et al.* Patient-derived Siglec-6-targeting antibodies engineered for T-cell recruitment have potential therapeutic utility in chronic lymphocytic leukemia. *J Immunother Cancer* **10** (2022). <https://doi.org/10.1136/jitc-2022-004850>

REVIEWERS' COMMENTS

Reviewer #1 (Remarks to the Author):

The authors have addressed all of my original comments in the submitted revision. In particular, the addition of the PDX mouse model data has improved the manuscript. The disappearance of the Siglec-6+ fraction from the original MEC-1 cells remains a mystery, and it seems unlikely that this is due to differences in passage number. The lack of difference in Siglec-6 expression between CXCR4^{lo}CD5^{hi} and CXCR4^{hi}CD5^{lo} fractions is intriguing. As is the lack of association between Siglec-6 and CD49d expression. Both of these findings perhaps suggest that the role of Siglec-6 in CLL is confined to the bone marrow microenvironment rather than the lymphoid tissues. In this regard, it would be fascinating to know whether CLL patients who are siglec-6+ have higher levels of residual disease following treatment.

Reviewer #2 (Remarks to the Author):

I thank the authors for addressing the point I raised. In my opinion this has strengthened the manuscript and I have no further comments.

Reviewer #3 (Remarks to the Author):

The authors addressed most comments

NCOMMS-23-03668A

Siglec-6 as a therapeutic target of cell migration and adhesion in Chronic Lymphocytic Leukemia

Reviewer comments are highlighted in italics.

Reviewer 1:

The authors have addressed all of my original comments in the submitted revision. In particular, the addition of the PDX mouse model data has improved the manuscript. The disappearance of the Siglec-6+ fraction from the original MEC-1 cells remains a mystery, and it seems unlikely that this is due to differences in passage number. The lack of difference in Siglec-6 expression between CXCR4loCD5hi and CXCR4hiCD5lo fractions is intriguing. As is the lack of association between Siglec-6 and CD49d expression. Both of these findings perhaps suggest that the role of Siglec-6 in CLL is confined to the bone marrow microenvironment rather than the lymphoid tissues. In this regard, it would be fascinating to know whether CLL patients who are siglec-6+ have higher levels of residual disease following treatment.

Response: We thank the reviewer for bringing up these interesting points. We agree that the lack of difference in Siglec-6 expression between CXCR4loCD5hi and CXCR4hiCD5lo fractions as well as lack of association between Siglec-6 and CD49d expression suggests an organ specific role of Siglec-6 in the bone marrow microenvironment. The status of residual disease in CLL patients who are Siglec-6+ is worth exploring, because it may serve as a prognosis factor, and will help to determine the best treatment strategy for CLL patients. Future studies in the lab will be focused on evaluating Siglec-6 expression and function in patients with aggressive CLL including Richter's transformation.

Reviewer 2:

I thank the authors for addressing the point I raised. In my opinion this has strengthened the manuscript and I have no further comments.

Response: We thank the reviewer for their feedback and critique which helped strengthen the manuscript.

Reviewer 3:

The authors addressed most comments.

Response: We thank the reviewer for bringing up interesting points in the context of Siglec-6 targeted therapeutics that are worth investigating in future studies.